# Stable Gastric Pentadecapeptide BPC 157 May Counteract Myocardial Infarction Induced by Isoprenaline in Rats

**DOI:** 10.3390/biomedicines10020265

**Published:** 2022-01-26

**Authors:** Ivan Barisic, Diana Balenovic, Mario Udovicic, Darija Bardak, Dean Strinic, Josipa Vlainić, Hrvoje Vranes, Ivan Maria Smoday, Ivan Krezic, Marija Milavic, Suncana Sikiric, Sandra Uzun, Gordana Zivanovic Posilovic, Sanja Strbe, Ivan Vukoja, Eva Lovric, Marin Lozic, Marko Sever, Martina Lovric Bencic, Alenka Boban Blagaic, Anita Skrtic, Sven Seiwerth, Predrag Sikiric

**Affiliations:** 1Department of Pharmacology, School of Medicine, University of Zagreb, 10000 Zagreb, Croatia; inbarisic@gmail.com (I.B.); diana.balenovic@gmail.com (D.B.); darija.bardak@gmail.com (D.B.); destrinic@gmail.com (D.S.); hrvoje.vranes@gmail.com (H.V.); rvansmodayl1@gmail.com (I.M.S.); ivank.ezic94@gmail.com (I.K.); gordana1709@gmail.com (G.Z.P.); strbes@gmail.com (S.S.); abblagaic@mef.hr (A.B.B.); 2Department of Internal Medicine, School of Medicine, University of Zagreb, 10000 Zagreb, Croatia; marioudovicic@gmail.com (M.U.); miminamama@gmail.com (M.L.B.); 3Laboratory for Advanced Genomics, Division of Molecular Medicine, lnstitute Ruder Boskovic, 10000 Zagreb, Croatia; josipa.vlainic@irb.hr; 4Department of Pathology, School of Medicine, University of Zagreb, 10000 Zagreb, Croatia; marija.milavic@mef.hr (M.M.); suncanasikiric@gmail.com (S.S.); eva.lovric@kb-merkur.hr (E.L.); sven.seiwerth@mef.hr (S.S.); 5Clinic of Anaesthesiology, Reanimatology and Intensive Care Zagreb, University Hospital Centre Zagreb, 10000 Zagreb, Croatia; sandra.uzun@vip.hr; 6School of Medicine, University of Osijek, 31000 Osijek, Croatia; iv.vukoja@gmail.com; 7Department of Pediatric and Preventive Dentistry, School of Dental Medicine, University of Zagreb, 10000 Zagreb, Croatia; marin.lozic@yahoo.com; 8Department of Surgery, School of Medicine, University of Zagreb, 10000 Zagreb, Croatia; dr.sever.marko@gmail.com

**Keywords:** pentadecapeptide BPC 157, isoprenaline, occlusion-like syndrome, myocardial infarction, oxidative stress, L-NAME, L-arginine, rats

## Abstract

We revealed that the stable gastric pentadecapeptide BPC 157, a useful peptide therapy against isoprenaline myocardial infarction, as well as against isoprenaline myocardial reinfarction, may follow the counteraction of the recently described occlusion-like syndrome, induced peripherally and centrally, which was described for the first time in isoprenaline-treated rats. BPC 157 (10 ng/kg, 10 µg/kg i.p.), L-NAME (5 mg/kg i.p.), and L-arginine (200 mg/kg i.p.) were given alone or together at (i) 30 min before or, alternatively, (ii) at 5 min after isoprenaline (75 or 150 mg/kg s.c.). At 30 min after isoprenaline 75 mg/kg s.c., we noted an early multiorgan failure (brain, heart, lung, liver, kidney and gastrointestinal lesions), thrombosis, intracranial (superior sagittal sinus) hypertension, portal and caval hypertension, and aortal hypotension, in its full presentation (or attenuated by BPC 157 therapy (given at 5 min after isoprenaline) via activation of the azygos vein). Further, we studied isoprenaline (75 or 150 mg/kg s.c.) myocardial infarction (1 challenge) and reinfarction (isoprenaline at 0 h and 24 h, 2 challenges) in rats (assessed at the end of the subsequent 24 h period). BPC 157 reduced levels of all necrosis markers, CK, CK-MB, LDH, and cTnT, and attenuated gross (no visible infarcted area) and histological damage, ECG (no ST-T ischemic changes), and echocardiography (preservation of systolic left ventricular function) damage induced by isoprenaline. Its effect was associated with a significant decrease in oxidative stress parameters and likely maintained NO system function, providing that BPC 157 interacted with eNOS and COX2 gene expression in a particular way and counteracted the noxious effect of the NOS-blocker, L-NAME.

## 1. Introduction

Isoprenaline myocardial infarction in rats is known to be a rapid, simple and non-invasive method that produces myocardial damage similar to that seen in acute myocardial infarction in humans [1]. We hypothesize that the stable gastric pentadecapeptide BPC 157 (for review see, i.e., [2,3,4]) may be a useful peptide therapy against isoprenaline-induced myocardial infarction and reinfarction as well as against isoprenaline-induced antecedent early noxious effects, which are, so far, less well-investigated.

Considering the particular points of the isoprenaline application [1], this may be because BPC 157 is known for its particular beneficial effect in congestive heart failure [5] and in many severe arrhythmias models [6,7,8,9,10,11]. It also demonstrates a therapeutic effect in pulmonary hypertension studies [12,13]. Moreover, there is a rapid effect of BPC 157 therapy on heart disturbances in studies of the permanent occlusion of major vessels-induced occlusion syndromes [14,15,16,17,18,19,20], “occlusion-like” syndrome induced by severe intoxication (alcohol, lithium) [21,22] and intra-abdominal hypertension [23], recovered with the activation of the collateral pathways to compensate vascular failure [14,15,16,17,18,19,20,21,22,23]. In these instances, there is compelling evidence that BPC 157 therapy exerted both a prophylactic and curative effect. This beneficial effect may be likely extended to the attenuation of the isoprenaline-induced noxious course. This may also be because catecholamines might represent a common pathway in the evolution of myocardial changes in humans who develop myocardial lesions without the narrowing or obstruction of coronary arteries [24].

For illustration, an activated azygos vein as a rescuing pathway, avoiding both the lung and liver, combines the inferior caval vein and superior caval vein via direct blood delivery. Thus, an activated azygos vein shunt could reorganize blood flow and instantly attenuate the consequences of maintained occlusion-induced vascular failure, both peripherally and centrally [14,15,16,17,18,19,20]. With major vascular occlusions, there was, at the periphery, the leading role of the rapid heart disturbances, and thereby, heart, lung, liver, kidney and gastrointestinal lesions, inferior and superior vena caval congestion, azygos vein failure, portal and caval hypertension and aortal hypotension occur [14,15,16,17,18,19,20]. Centrally, there was intracranial (superior sagittal sinus) hypertension, brain swelling and severe lesions [14,15,16,17,18,19,20]. Widespread thrombosis occurred peripherally and centrally (note that BPC 157 maintained thrombocyte function (without interference with coagulation) [25,26,27], prevented thrombosis formation in veins and arteries, and abrogated already advanced thrombosis [14,15,16,17,18,19,20,21,22,23,28]). A shared major “occlusion-like” syndrome, which was also antagonized by BPC 157 therapy, occurred after intragastric application of absolute alcohol, intraperitoneal application of lithium overdose, or with maintained severe intra-abdominal hypertension, grade III and grade IV [21,22,23]. Thus, the demonstrated leading role of the rapid heart disturbances and general vascular failure in the major vessels-induced occlusive syndromes [14,15,16,17,18,19,20], severe intoxication (i.e., alcohol, lithium) [21,22] and maintained intra-abdominal hypertension [23] suggest that an application of isoprenaline by its own can induce rapidly its own “occlusion-like”-syndrome, an early disturbance decisive for further isoprenaline-myocardial lesion progression. This particular early noxious effect of isoprenaline has not been investigated so far, but it can be likely antagonized by BPC 157 therapy application.

Isoprenaline-induced lesions appeared due to damaging free radical induction [29]. BPC 157 acts as a stabilizer of cellular junctions [30] and free radical scavenger [31,32,33]. It was found to counteract free radical formation and lesions, in particular, in vascular occlusion studies [14,15,16,17,18,19,34,35,36,37]. Therefore, this point will be particularly investigated.

In the present study, isoprenaline-treated rats received pentadecapeptide BPC 157, L-NAME, and L-arginine alone and/or in combination. Further assessments included eNOS, iNOS, COX-2 mRNA levels and lipid peroxidation in infarcted hearts (known to be up-regulated in isoprenaline myocardial infarction [38,39,40]). Namely, isoprenaline interacts with the NO system and its blockade [38,41,42,43]. BPC 157/NO system relations illustrate that BPC 157 is known to affect several molecular pathways [30,44,45,46,47,48,49,50,51,52], in particular, having modulatory effects on the NO system and prostaglandin system [53,54], on vasomotor tone, and on the activation of the Src-Caveolin-1-eNOS pathway [45]. Consequently, BPC 157 induced NO release of its own [55,56], and was shown to have a large interaction with the NO system in various models and species [53]. It also demonstrated the counteraction of the adverse effects of either NOS blockade (L-NAME) or NOS substrate L-arginine application (i.e., it opposed hypertension and pro-thrombotic effects (L-NAME) [26,55], as well as hypotension and anti-thrombotic (L-arginine) effects [26,55]). Furthermore, in the isoprenaline-treated rats, the implementation of NO system blockade (L-NAME), NO system over-stimulation (L-arginine) and NO system immobilization (L-NAME + L-arginine) would fully consider the role of the NO system as an endogenous cardioprotectant [38] (i.e., in rats and mice, doxorubicin congestive heart failure therapy with BPC 157 resulted in the normalization of NO system functioning due to the induced normalization of the increased endothelin-1 values [5]).

We also included investigations related to the necrosis markers commonly measured in routine practice [57], including serum creatine kinase (CK), creatine kinase myocardial band specific to cardiomyocytes (CK-MB), LDH, and troponin T (cTnT).

## 2. Materials and Methods

### 2.1. Animals

Male Wistar Albino rats (200–250 g) were used in all experiments and were randomly assigned, with 10 rats per experimental group, were used. Rats were fed with a standard rodent chow ad libitum. All experiments were approved by the local ethics committee.

### 2.2. Drugs and Materials

Pentadecapeptide BPC 157 (GEPPPGKPADDAGLV, M.W. 1419), (Diagen, Ljubljana, Slovenia), dissolved in saline, was used in all experiments. The peptide BPC 157 is part of the sequence of human gastric juice protein BPC and is freely soluble in water at pH 7.0 and saline [2,3,4]. It was prepared as described previously with 99% high-pressure liquid chromatography (HPLC) purity, expressing 1-des-Gly peptide as an impurity (for review, see, i.e., [14,15,16,17,18,19,20,21,22,23]). Isoprenaline (isoprenaline hydrochloride, Sigma, St. Louis, MO, USA), L-NAME (Sigma, St. Louis, MO, USA) and L-arginine (Sigma, St. Louis, MO, USA) were accordingly used [24,53].

### 2.3. Noxious Procedure, Medication and Assessment

The assessment was conducted by measuring ECG recordings, serum enzymes, troponin T, RNA extraction and RT-PCR (GAPDH, iNOS, eNOS and COX-2 mRNA levels genes), macro/microscopic lesion presentation, clinical presentation, echocardiography and oxidative stress. For the early isoprenaline-induced occlusion-like syndrome, we also assessed multiorgan failure (brain, heart, lung, liver, kidney and gastrointestinal lesions), thrombosis, intracranial (superior sagittal sinus) hypertension, portal and caval hypertension, and aortal hypotension.

We induced myocardial isoprenaline lesions (as described [24] with one or two applications of isoprenaline (75 mg/kg or 150 mg/kg s.c.)). The isoprenaline application was once (at time 0) for initial infarct induction, assessed at 24 h thereafter; then, at the point of 24 h, the next application was given (isoprenaline given twice, at 0 h and 24 h) for re-infarct induction. Assessment was again carried out at the end of the subsequent 24 h period. Medication (BPC 157 (10 ng/kg, 10 µg/kg i.p.), L-NAME (5 mg/kg i.p.), and L-arginine (200 mg/kg i.p.) given alone or together) was administered at (i) 30 min before isoprenaline, or, alternatively, (ii) at 5 min after isoprenaline (75 mg/kg s.c. or 150 mg/kg s.c.) on day 1 and day 2. In combination studies, we used a higher isoprenaline dose, BPC 157 10 µg/kg i.p., L-NAME and/or L-arginine, given alone and/or combined. Controls simultaneously received an equivolume of saline (5.0 mL/kg i.p.).

At 30 min after isoprenaline 75 mg/kg s.c. and medication (BPC 157 (10 ng/kg, 10 µg/kg i.p. or saline 5.0 mL/kg i.p.) given at 5 min after isoprenaline, we assessed the early “occlusion-like” syndrome in the isoprenaline-treated rats as an early multiorgan failure (brain, heart, lung, liver, kidney and gastrointestinal lesions), thrombosis, intracranial (superior sagittal sinus) hypertension, portal and caval hypertension, and aortal hypotension, full presentation.

### 2.4. Gross Lesion Presentation

The gross lesions were recorded in deeply anaesthetized, laparatomized, craniotomized and thoracotomized rats, with a camera attached to a VMS-004 Discovery Deluxe USB microscope (Veho, Dayton, OH, USA) for the early “occlusion-like” syndrome assessment. Features investigated in recorded photographs included the hemorrhagic lesions in the stomach assessed as the sum of the longest lesion diameters after isoprenaline administration and following medication at 30 min; the rats were then sacrificed. The other investigated features included the gross presentation of the brain, the superior mesenteric vein, the inferior vena cava, azygos vein and the heart assessment.

For the group which underwent infarction and re-infarction induction at 24 h after first isoprenaline, at the point of 24 h after the second application (isoprenaline given twice, at 0 h and 24 h), the rat’s chest cavities were opened to remove the heart shortly after blood samples were taken. Hearts were assessed for lesions expressed as percentage of the injured area.

### 2.5. Assessment of the Change in the Brain and Veins’ Volume Proportional to the Change in the Brain and Veins’ Surface Area

As described previously [17,18,19,20,21,22,23], the presentation of the brain (in vivo and ex vivo) and peripheral veins (superior mesenteric vein, inferior vena cava and azygos vein) was recorded in deeply anaesthetized rats that had undergone laparotomy or complete calvariectomy using a camera attached to a VMS-004 Discovery Deluxe USB microscope (Veho, Dayton, OH, USA). This endeavor was performed in every rat group after application of isoprenaline and after application of therapy (BPC 157 or saline) both; the rats were then sacrificed. The border of the brain or veins in photographs was marked using ImageJ software (National Institutes of Health, Bethesda, MD, USA). Next, the surface area (in pixels) of the brain or veins was measured using a measuring function. This was performed with the brain or vein photographs for each of the above-mentioned groups for both control and BPC 157 treated animals. Then, after surfaces of the brain and veins were measured, the saline-to-BPC 157 surface ratio of the two surfaces was calculated as ($\frac{A_{2}}{A_{1}}$), where *A*_2_ is the surface area of the brain (or vein) in saline-treated animals, and *A*_1_ is the surface area of the brain (or veins) in BPC 157-treated animals. Starting from the square-cube law, Equations (1) and (2), an equation for the saline-to-BPC 157 brain (or vein) volume ratio proportional to the the saline-to-BPC 157 brain (or vein) surface area ratio Equation (6) was derived. For Equations (1)–(5), any arbitrary one-dimensional length on the photograph was defined (e.g., the rostro-caudal length of the brain, or any arbitrary length of a vein). This was used only to define the one-dimensional proportion (*l*_2_*/l*_1_) between two observed brains or veins and as an inter-factor (and because of that, not measured) for deriving the final Equation (6). The procedure was as follows:(1)$$A_{2}=A_{1}\times {\left(\frac{l_{2}}{l_{1}}\right)}^{2}$$

(1) Square-cube law;


(2)
$$V_{2}=V_{1}\times {\left(\frac{l_{2}}{l_{1}}\right)}^{3}$$


(2) Square-cube law;


(3)
$$\frac{A_{2}}{A_{1}}={\left(\frac{l_{2}}{l_{1}}\right)}^{2}$$


(3) From (1), after dividing both sides by *A*_1_;


(4)
$$\frac{l_{2}}{l_{1}}=\sqrt{\frac{A_{2}}{A_{1}}}$$


(4) From (3), after taking the square root of both sides;


(5)
$$\frac{V_{2}}{V_{1}}={\left(\frac{l_{2}}{l_{1}}\right)}^{3}$$


(5) From (2), after dividing both sides by *V*_1_;


(6)
$$\frac{V_{2}}{V_{1}}={\left(\sqrt{\frac{A_{2}}{A_{1}}}\right)}^{3}$$


(6) After incorporating (4) into (5).

This measuring procedure, followed by the calculation of volume ratios, was performed separately for the brain and veins.

Brain swelling was recorded in separate rats 15 min after complete calvariectomy. Briefly, six burr holes were drilled in three horizontal lines, all of them medial to the superior temporal lines and temporalis muscle attachments. The two rostral burr holes were placed just basal from the posterior interocular line, the two basal burr holes were placed just rostral to the lambdoid suture (and transverse sinuses) on both sides, respectively, and the middle two burr holes were placed in the line between the basal and rostral burr holes.

### 2.6. Thrombus Assessment

After euthanasia at 30 min after isoprenaline, the superior sagittal sinus, the portal vein, the inferior vena cava, the superior mesenteric artery, and the abdominal aorta were removed from the rats, and clots were weighed [14,15,16,17,18,19,20,21,22,23].

### 2.7. Superior Sagittal Sinus, Portal Vein, Vena Caval and Abdominal Aortal Pressure Recording

Recordings were made in deeply anaesthetized rats with a cannula (BD Neoflon Cannula) connected to a pressure transducer (78534C MONITOR/TERMINAL; Hewlett Packard, Palo Alto, CA, USA) inserted into the superior sagittal sinus, portal vein, inferior vena cava and abdominal aorta at the level of the bifurcation at 30 min after isoprenaline 75 mg/kg sc (providing saline or BPC 157 given at 5 min after isoprenaline); the rats were then sacrificed. Each recording lasted 1 min. For the superior sagittal sinus pressure recording, we made a single burr hole in the rostral part of the sagittal suture, above the superior sagittal sinus, and cannulated the anterior portion of the superior sagittal sinus by Braun intravenous cannulas. We then laparatomized the rats for the portal vein, inferior vena caval and abdominal aortal pressure recordings.

Of note, normal rats exhibited a superior sagittal sinus pressure of −24 to −27 mmHg and a portal pressure of 3–5 mmHg, similar to that of the inferior vena cava, although this was at least 1 mmHg higher in the portal vein. By contrast, the abdominal aorta blood pressure was 100–120 mm Hg at the level of the bifurcation [14,15,16,17,18,19,20,21,22,23].

### 2.8. Microscopy

#### 2.8.1. Tissue Preparation

Organ tissue was fixed in 10% neutral buffered formalin (pH 7.4) at room temperature for 24 h (brain, liver, kidney, lungs, heart, stomach, intestines). Representative tissue specimens were collected, dehydrated and embedded in paraffin blocks, sectioned at 4 μm, and stained with hematoxylin-eosin according to the following automated Sakura Tissue-Tek DRS 2000 Slide Stainer protocol (https://www.sakura.eu/Solutions/Staining-Coverslipping/H-E-Kit accessed on 19 October 2021): rehydration in distilled water, staining with hematoxylin, washing in running tap water, differentiation with 70% alcohol, staining with eosin, dehydration, clearing, and mounting.

Tissue specimens were evaluated microscopically by two examiners (board-certified pathologists, A.S. and E.L.) based on blind assessment. Histological examination was conducted using an Olympus BX51 microscope, and images of tissue were saved as uncompressed 24-bit RGB TIFF files using an Olympus 71 digital camera.

#### 2.8.2. Brain Histology

In the assessment of the “occlusion-like” syndrome in isoprenaline-treated rats sacrificed at 30 min after isoprenaline 75 mg/kg s.c. (providing saline or BPC 157 given at 5 min after isoprenaline), the brain was dissected according to NTP-7, at Level 3 and 6 with neuroanatomic subsites presented in certain brain sections using coronal sections with 3 mandatory sections and analyzed using a semiquantitative neuropathological scoring system, as previously described [16,17,18,19,20,21,22,58], and combined score (0–8) = sum of analyzed affected areas (0–4) and karyopyknotic cells in the brain areas (0–4), as follows. Specifically, we analyzed affected brain areas (0–4), cerebral (NTP-7, Level 3), cerebellar cortex (NTP-7, Level 6), and the hippocampus, thalamus, and hypothalamus (NTP-7, Level 3) as follows (score 0 indicates no histopathologic change): score 1: small, patchy, complete or incomplete infarcts (≤10% of area affected); score 2: partly confluent or incomplete infarcts (20–30% of area affected); score 3: large confluent complete infarcts (40–60% of area affected); score 4: in cortex, total disintegration of the tissue; in hypothalamus, thalamus, and hippocampus large complete infarcts (˃75% of area affected). Karyopyknotic cells in the affected brain areas (0–4), cerebral (NTP-7, Level 3), cerebellar cortex (NTP-7, Level 6), hippocampus, thalamus, and hypothalamus (NTP-7, Level 3) were analyzed as follows (score 0 indicates no change): score 1: a few karyopyknotic of neuronal cells (≤20%); score 2: patchy areas of karyopyknotic cells (50%); score 3: more extensive karyopyknotic areas (75%); score 4: complete infarction (100%).

#### 2.8.3. Lung Histology

In the assessment of the “occlusion-like” syndrome in isoprenaline-treated rats sacrificed at 30 min after administration of isoprenaline 75 mg/kg s.c. (providing saline or BPC 157 given at 5 min after isoprenaline), we used a scoring system to grade the degree of lung injury, including observations of focal thickening of the alveolar membranes, congestion, pulmonary edema, intra-alveolar hemorrhage, interstitial neutrophil infiltration and intra-alveolar neutrophil infiltration. Each feature was assigned a score from 0 to 3 based on its absence (0) or presence to a mild (1), moderate (2) or severe (3) degree. A total cumulative histology score was determined [21,22,23,59].

#### 2.8.4. Renal, Liver and Heart Histology

The renal injury was based on the degeneration of the Bowman’s space, glomeruli and proximal and distal tubules; vascular congestion; and interstitial edema. The criteria for liver injury were vacuolization of hepatocytes and pyknotic hepatocyte nuclei, activation of Kupffer cells and enlargement of sinusoids. Each specimen was scored using a scale from 0 to 3 (0, none; 1, mild; 2, moderate; 3, severe) for each criterion [21,22,23,60].

The myocardium was graded for the severity of necrosis exhibited in the ventricles. The mean value of their scores is presented. The pathological criteria for grading the severity of necrosis were: score 1 (mild), one or two small foci; score 2 (slight), several small foci; score 3 (moderate), multiple small foci or several large foci; score 4 (severe), multiple large foci or diffuse area of necrosis [21,22,23,61].

#### 2.8.5. Gastrointestinal Histology

Intestinal tissue damage was analyzed using a histologic scoring scale adapted from Chui and co-workers [62] on a scale from 0 to 5 (normal to severe) in 3 categories (mucosal injury, inflammation, hyperemia/hemorrhage) for a total score of 0 to 15 as described by Lane and co-workers [63]. Morphologic features of mucosal injury were based on different grades of epithelia lifting, villi denudation, and necrosis; grades of inflammation were from focal to diffuse according to lamina propria infiltration or subendothelial infiltration; hyperemia/hemorrhage was graded from focal to diffuse according to lamina propria or subendothelial localisation.

### 2.9. ECG Recording

In deeply anesthetized rats, an ECG was recorded continuously in all three main leads by positioning stainless steel electrodes on all four limbs using an ECG monitor by 2090 Medtronic programmer (Minneapolis, MN, USA) connected to a digital oscilloscope LeCroywaverunnerLT342 (Chestnut Ridge, NY, USA), which enabled precise recordings, measurements and analysis of ECG parameters. At 30 min, 24 h and 48 h, the ECG changes were analyzed [14,15,16,17,18,19,20,21,22,23].

### 2.10. Biochemical Analysis

Serum enzymes (creatine phosphokinase, CK; myocardial fraction, MBCK; and lactatate dehydrogenase, LDH) were measured using an auto-analyzer AU-800 (Olympus, Tokyo), and troponin T was assessed by a rapid quantitative assay using the Cardiac Reader (Roche Diagnostics, Mannheim, Germany).

### 2.11. RNA Extraction and RT-PCR

RNA was extracted from 5 to 10 µm thick sections of paraffin-embedded heart septum tissue using the High Pure RNA Paraffin Kit (Roche, Basel, Switzerland). RNA density was measured using NanoDrop ND-1000 Spectrophotometer (NanoDrop Technologies, Wilmington, DE, USA). A total of 150 ng of RNA from each sample was used in RT-PCRs performed in a total volume of 25 µL using the QIAGEN OneStep RT-PCR Kit (Qiagen, Venlo, The Netherlands). The RT-PCR mixture contained 5 µL 5xQIAGEN OneStep RT-PCR Buffer, 1 µL dNTP Mix, 1.5 µL of 10 µM sense primer, 1.5 µL of 10 µM antisense primer, 1 µL OneStep RT-PCR Enzyme Mix, RNA as described and RNase-free water. RT-PCR was performed in a thermocycler GeneAmp PCR System 9700 (Applied Biosystems, Foster City, CA, USA). The samples were incubated at 55 °C for 30 min to enable reverse transcription, followed by heating at 94 °C for 15 min to activate the Hot StartTaq DNA Polymerase. After that, the samples were incubated at 94 °C for 30 s, 55 °C for 30 s and 72 °C for 30 s for 40 cycles with GAPDH, iNOS, eNOS and COX-2 primers (Table 1). At the end, samples were incubated for 10 min at 72 °C. Results were expressed as the ratio of the optical density of the NOS PCR or COX-2 PCR products to the density of the corresponding GAPDH PCR products [64].

### 2.12. Gross Clinical Assessment

Assessment included edema (involving the face, upper and lower lip, snout, paws and scrotum (presented with extreme cyanosis)) that may be prominent, scored from 0 to 3 (0—without swelling, 1—mild swelling, 2—moderate swelling and 3—severe swelling), poor respiration as respiratory rate [5] and the number of fatalities after one or two applications of isoprenaline.

### 2.13. Echocardiography

Echocardiography (2D) was performed in all animals 24 h and 48 h after isoprenaline (75 and 150 mg/kg s.c. for the second time at 0 and 24 h) with a 7.5-MHz, phased-array transducer (SONOS 2500 Hewlett Packard). The image of the left ventricular cavity was obtained from a long parasternal view, and the M mode cursor was positioned between the papillary muscles, just below the mitral valve plane. M-mode measurements of the LV cavity size in systole and diastole were made, and global LV function was assessed by calculating ejection fraction (EF) from end-diastolic and end-systolic volumes of the left ventricle according to the following equations: diastolic volume (DV) (mL) = 0.001047 × left ventricle internal dimension in diastole (LVIDd (mm)) (Balenovic et al., 2009); systolic volume (SV) (mL) = 0.001047 × left ventricle internal dimension in systole (LVIDs (mm)) [8]; stroke volume (mL) = DV-SV; ejection fraction of left ventricle (EF LV) (%) = stroke volume (mL)/DV (mL) × 100 (in healthy animals, DV 0.16 ± 0.03, SV 0.013 ± 0.002, EF LV 90 ± 4). All measurements were made in accordance with the recommendations of the European Association of Echocardiography and the American Society of Echocardiography [65].

### 2.14. Oxidative Stress

At the end of the 24 h period in isoprenaline-treated rats that received a higher dose of BPC 157 before or after the isoprenaline challenge, oxidative stress in the collected tissue samples was assessed by quantifying thiobarbituric acid-reactive species (TBARS) as malonedialdehyde (MDA) equivalents. The tissue samples were homogenized in PBS (pH 7.4) containing 0.1 mM butylated hydroxytoluene (BHT) (TissueRuptor, Qiagen, Valencia, CA, USA) and sonicated for 30 sec in an ice bath (Ultrasonic bath, Branson, MO, USA). Trichloroacetic acid (TCA, 10%) was added to the homogenate, the mixture was centrifuged at 3000 rpm for 5 min, and the supernatant was collected. Then, 1% TBA was added, and the samples were boiled (95 °C, 60 min). The tubes were then kept on ice for 10 min. Following centrifugation (14,000 rpm, 10 min), the absorbance of the mixture at the wavelength of 532 nm was determined. The concentration of MDA was read from a standard calibration curve plotted using 1,1,3,3′-tetraethoxy propane (TEP). The extent of lipid peroxidation was expressed as MDA using a molar extinction coefficient for MDA of 1.56 × 105 mol/L/cm. The protein concentration was determined using a commercial kit. The results are expressed in nmol per mg of protein.

### 2.15. Statistical Analysis

Statistical analysis was performed using parametric two-way mixed model ANOVA (one factor is repeated-measures) and Student–Newman–Keuls test to compare the difference between groups. Fisher’s exact probability test was used to assess the number of dead and surviving rats. A *p* value of 0.05 or less was considered statistically significant.

## 3. Results

To clarify the isoprenaline infarction/reinfarction and stable gastric pentadecapeptide BPC 157, we challenge the possibility that the stable gastric pentadecapeptide BPC 157, as a useful peptide therapy against isoprenaline myocardial infarction, as well as against isoprenaline myocardial reinfarction, should follow the activation of the collateral rescuing pathways seeable for the counteraction of the recently described occlusion-like syndrome, induced peripherally and centrally, that would also appear in the isoprenaline-treatedrats, being decisive for further full myocardial infarction development. At 30 min after isoprenaline 75 mg/kg s.c., we noted an early multiorgan failure (brain, heart, lung, liver, kidney and gastrointestinal lesions), thrombosis, intracranial (superior sagittal sinus) hypertension, portal and caval hypertension, and aortal hypotension, full presentation (or attenuation by BPC 157 therapy (given at 5 min after isoprenaline) via activation of the azygos vein) (Figure 1, Figure 2, Figure 3, Figure 4, Figure 5, Figure 6, Figure 7, Figure 8 and Figure 9).

In further course, isoprenaline challenge(s) resulted in uniform severe myocardial lesions presentation (Figure 10, Figure 11, Figure 12, Figure 13, Figure 14, Figure 15, Figure 16, Figure 17, Figure 18 and Figure 19, Table 2, Table 3 and Table 4). Contrarily, the beneficial effects of BPC 157 appeared with either a prophylactic or therapeutic regimen as assessed by all assessments. We also noted particular effect of the application of the NO agents NOS-blocker L-NAME and NOS-substrate L-arginine given alone and/or in combination. Additionally, we noted a particular effect of BPC 157 when given with NO agents.

### 3.1. Early Occlusion-Like Syndrome and BPC 157 Therapy

At 30 min after isoprenaline 75 mg/kg s.c., as an occlusion-like syndrome, hearts regularly presented in isoprenaline control rats with congested blood vessels of the myocardium with areas of numerous subendocardial ishemic myocytes, while BPC 157 rats showed no congestion in the heart, only some subendocardial ischemic myocytes and normal T-wave (Figure 1, Figure 3 and Figure 4).

On ECG, isoprenaline control rats had giant T-waves (0.5 ± 0.02 mV), while BPC 157 rats showed normal T-waves (0.2 ± 0.02 (µg) 0.2 ± 0.03 (ng) mV) (*p* ˂ 0.05, vs. control, at least).

Isoprenaline control rats exhibited congestion and dilatation of medium and small blood vessels of the lung parenchyma. Contrarily, BPC 157-treated rats had no change or only mild congestion of the capillaries in the lung parenchyma.

Isoprenaline control rats demonstrated marked dilatation and congestion of blood vessels in the portal tracts of the liver, as well as dilatation of central veins and sinusoids in the liver parenchyma. Contrarily, no changes were found in the liver tissue in BPC 157-treated rats (Figure 1 and Figure 5).

Isoprenaline control rats exhibited conspicuous dilatation of blood vessels and congestion in the kidney tissue, as well as glomeruli. Contrarily, BPC 157-treated rats had only mild dilatation and congestion of blood vessels in the kidney tissue (Figure 1 and Figure 5).

Isoprenaline control rats exhibited hemorrhagic gastric lesions (5.5 ± 1.0, means ± SD, mm, sum of the longest lesions diameters), while BPC 157-treated rats had spared gastric mucosa (*p* ˂ 0.05, vs. control, at least). An increased rate of congestion was found in the gastrointestinal tract of control animals, from the stomach to colon, while only mild congestion was found in the colon with no changes in the stomach and small intestine in BPC 157-treated rats (Figure 1 and Figure 6).

Grossly, BPC 157 therapy strongly antagonized brain swelling (Figure 1, Figure 3, Figure 7, Figure 8, Figure 9). Control rats exhibited pronounced edema and congestion in the brain tissue. Intracerebral hemorrhage involving larger areas of brain tissue was observed. Marked intracerebral bleeding was seen in the area of the corpus callosum, amygdala, thalamus, neocortex, and striatum. Intraventricular hemorrhage involving the third and lateral ventricles was also seen (Figure 7).

In contrast, in the BPC 157-treated rats, mild edema and congestion was visible in brain tissue. Intracerebral hemorrhage affected only the corpus callosum, and there was no intraventricular hemorrhage (Figure 7).

An increased number of karyopyknotic cells was found in all four regions: cerebral and cerebellar cortex, hippocampus cortex, and hypothalamus/thalamus in control rats (Figure 8 and Figure 9). In particular, marked karyopyknosis of pyramidal cells of the hippocampus was observed, followed by moderate karyopyknosis and degeneration of Purkinje cells of the cerebellar cortex. Contrarily, BPC 157-treated rats showed a few karyopyknotic neuronal cells in the analyzed neuroanatomic structures (Figure 8 and Figure 9).

At 30 min after isoprenaline treatment, we noted an early thrombosis, arterial and venous, peripherally and centrally. Of note, BPC 157 therapy markedly attenuated thrombosis both in arteries and veins, peripherally and centrally (Figure 1).

Regarding blood pressure, isoprenaline control rats exhibited intracranial (superior sagittal sinus) hypertension, portal and caval hypertension, and aortal hypotension. BPC 157 completely eliminated portal and caval hypertension, and markedly attenuated intracranial (superior sagittal sinus) hypertension and aortal hypotension (Figure 1).

As previously reported, BPC 157 therapy (given at 5 min after isoprenaline) goes via activation of the azygos vein (Figure 1 and Figure 2). Of note, presenting the portal and caval hypertension and organ congestion, it is reasonable that in the isoprenaline control rats, the azygos vein appears to be collapsed, and the superior mesenteric vein and inferior caval vein are apparently congested (Figure 1 and Figure 2). As a part of the defensive response with the BPC 157 treatment, these vessels’ presentation approaches the presentation of the normal vessels. Thus, an activated direct communication between the inferior and superior (left) caval vein through the azygos vein, thereby ensuring direct blood flow delivery, appears to be essential for blood flow restoration.

In conclusion, we established in the early isoprenaline course an early multiorgan failure (brain, heart, lung, liver, kidney and gastrointestinal lesions), thrombosis, intracranial (superior sagittal sinus) hypertension, portal and caval hypertension, and aortal hypotension versus attenuation by BPC 157 therapy (given at 5 min after isoprenaline) via activation of the azygos vein. This may be essential for further isoprenaline challenge(s) resulting in the presentation of uniform severe myocardial lesions and for BPC 157’s counteracting potential.

### 3.2. BPC 157 Therapy, Myocardial Infarction, and Reinfarction

Specifically, in BPC 157-treated rats, biochemical assessment demonstrated decreased cardiospecific serum CK-MB, CK, LDH and troponin T (Figure 10 and Figure 11).

BPC 157-treated rats showed evidently improved clinical presentation (close to healthy) (Table 2) and markedly improved macro (Figure 12 and Figure 14)/microscopic lesion presentation (Figure 13 and Figure 15).

In BPC 157-treated rats, ECG recordings exhibit reduced ST elevation/depression (Figure 6 and Figure 7). Additionally, there is reduced eNOS and COX-2 gene expression (Table 4).

In particular, in BPC 157-treated rats, echocardiography presentation shows better preserved heart function, as shown by improved DV, SV, EF LV, using both the prophylactic and therapeutic regimen (Figure 18 and Figure 19).

Indicatively, in BPC 157-treated rats, there is markedly attenuated isoprenaline-induced oxidative stress (Figure 19).

Thus, it is evident that BPC 157 markedly counteracts myocardial isoprenaline lesions. This is the case if one isoprenaline application of 75 mg/kg or 150 mg/kg s.c. is given for initial infarct induction and assessed at 24 h thereafter. Likewise, the counteraction includes the myocardial lesions after 2 isoprenaline applications (i.e., re-infarction), if the next application for re-infarct induction occurs at the point of 24 h and assessment after the end of the subsequent 24 h period. Specifically, this includes both 10 ng and 10 µg regimens. A sustained effect (given at 30 min before isoprenaline) and a rapid effect (given at 5 min after isoprenaline) were evidenced for counteraction of both isoprenaline regimens (75 mg/kg s.c. and 150 mg/kg s.c.). The salutary effect of BPC 157 therapy (10 ng and 10 µg/kg in pretreatment and post-treatment regimens) was also observed in terms of mortality, which was significantly lower in BPC 157 groups (Table 3).

### 3.3. NO-Agents and BPC 157 Therapy

Considering the NO agents’ effects, L-arginine was also beneficial, while L-NAME application further aggravated isoprenaline lesions given either before or after isoprenaline challenge(s). Further, we considered the effects of their mutual interaction. L-arginine brings down aggravation by NOS-blockade with L-NAME to the control levels. Illustratively, L-NAME appears to be more sensitive to BPC 157 application, providing that BPC 157 nullifies the effect of L-NAME and brings all isoprenaline (+L-NAME) lesions markedly below control values (L-NAME + BPC 157 + isoprenaline; L-NAME + L-arginine + BPC 157 + isoprenaline; isoprenaline + L-NAME + BPC 157; isoprenaline + L-NAME + L-arginine + BPC 157). With L-arginine, BPC 157 maintained its original beneficial effect.

## 4. Discussion

The relevance of the isoprenaline-induced myocardial infarct is supported by the evidence that catecholamines represent a common pathway in the evolution of myocardial changes in humans who develop myocardial lesions without narrowing or obstruction of coronary arteries [24]. Recently, our approach was the BPC 157 therapy effect (for review see, i.e., [2,3,4]) in the study of the permanent occlusion of major vessels-induced occlusion syndromes [14,15,16,17,18,19,20], the “occlusion-like” syndrome induced by severe intoxication (alcohol, lithium) [21,22] or maintained severe intra-abdominal hypertension [23] recovered with the activation of the collateral pathways [14,15,16,17,18,19,20,21,22,23]. In the present study, we revealed that the stable gastric pentadecapeptide BPC 157, a useful peptide therapy against isoprenaline myocardial infarction, as well as against isoprenaline myocardial reinfarction, may be consequent to the counteraction of a similar early occlusion-like syndrome that isoprenaline induced peripherally and centrally, which is, so far, not described. Early multiorgan failure, thrombosis, intracranial (superior sagittal sinus) hypertension, portal and caval hypertension, and aortal hypotension, in its full presentation (i.e., giant T-wave) or attenuated by BPC 157 therapy (i.e., normal T-wave) may be decisive for further isoprenaline-lesion development, similar to that noted with major intoxication (i.e., alcohol, lithium) or major vessel occlusion [14,15,16,17,18,19,20,21,22,23]. In these vessels, occlusion syndrome and occlusion-like syndrome development and BPC 157 therapy, and the activation of the collateral pathways to mobilize trapped blood volume via the azygos vein was shown to play a particular role as a rescuing pathway [14,15,16,17,18,19,20,21,22,23]. The azygos vein appeared completely collapsed in isoprenaline-treated rats (note, the rat azygos vein partially resembles atrial myocardium [66]), but its activation by BPC 157 therapy may directly deliver blood flow, compensate vessel occlusion and reorganize blood flow [14,15,16,17,18,19,20,21,22,23].

The results demonstrated that BPC 157 caused a significant reduction in all of the necrosis markers commonly measured in routine practice (i.e., CK, CK-MB, LDH, and cTnT) and attenuated gross (no visible infarcted area) and histological damage, including ECG (no ST-T ischemic changes) and echocardiography (preservation of systolic left ventricular function) damage induced by isoprenaline. Its effect was associated with a significant decrease in oxidative stress parameters and likely maintained NO system function, providing that BPC 157 interacted with eNOS and COX2 gene expression, in a particular way and counteracted the noxious effect of the NOS-blocker, L-NAME.

Of note, as mentioned, BPC 157 may counteract the isoprenaline-induced damage since the very beginning (i.e., 30 min) as it markedly attenuated the early isoprenaline-induced occlusion-like syndrome. Indicatively, the demonstrated leading role of rapid heart disturbances and general vascular failure in the major vessels-induced occlusive syndromes [14,15,16,17,18,19,20] and in occlusion-like syndrome with major intoxication (alcohol, lithium) or maintained severe intra-abdominal hypertension [21,22,23] means in the isoprenaline-treated rats, isoprenaline was detected as an early syndrome of multiorgan failure, peripherally and centrally. As before [17,18,19,20,21,22,23], we noted brain swelling, edema, large intracerebral hemorrhage and intraventricular hemorrhage in the central and lateral ventricles and considerable lesions in all brain areas. Intracranial (superior sagittal sinus) hypertension also occurred due to the harmful inability to drain venous blood adequately for a given cerebral blood inflow without raising venous pressures, which suddenly causes venous and intracranial hypertension [20]. Additionally, as the closely interrelated increased pressure in the three body cavities was rapidly transmitted through the venous system [23] (and thereby, the venous system can be seen as a therapy target [14,15,16,17,18,19,20,21,22,23]), the brain swelling and lesions and increased pressure in the superior sagittal sinus may induce a comparable deadly syndrome [20]. They may be, peripherally, marked heart congestion and large areas of subendocardial ischemic (pyknotic) myocytes, lung (hemorrhage), liver congestion and kidney congestion, gastrointestinal lesions, portal and caval hypertension and aortal hypotension. Marked thrombosis, arterial and venous, peripherally and centrally, occur as prominent stasis and vessel failure progress (i.e., congested inferior caval vein and superior mesenteric vein, failed azygos vein unable to serve as a rescuing pathway with direct delivery to the superior caval vein [14,15,16,17,18,19,20,21,22,23]). Thus, we revealed an antecedent early disturbance decisive for further isoprenaline myocardial lesion progression. Thereby, it is notable that BPC 157 mitigated all of these disturbances, whether by a primary or secondary effect, i.e., no intraventricular hemorrhage and attenuated intracranial (superior sagittal) hypertension. In the heart, no congestion was found in BPC 157 rats, with rare single or few myocytes showing pyknosis, no portal or caval hypertension, and attenuated aortal hypotension. The presence of almost abrogated thrombosis, arterial and venous, peripherally and centrally, may signify reactivated blood flow. In this instance, inferior caval vein and superior mesenteric vein demonstrated normal vessel presentation and azygos vein is recovered to serve as a rescuing pathway with direct delivery to the superior caval vein. This may further ascertain the attenuated isoprenaline course (Figure 20).

It is possible that isoprenaline-induced myocardial infarction appeared as a NO system failure [38] that BPC 157 corrected. BPC 157 induces NO release of its own [55,56] and controls vasomotor tone through the activation of the Src-Caveolin-1-eNOS pathway [46], superseding that of L-arginine. Namely, the beneficial effect of L-arginine brings down aggravation by NOS-blockade with L-NAME to the control levels. BPC 157 nullifies the effect of L-NAME and brings all isoprenaline (+L-NAME) lesions markedly below control values (L-NAME + BPC 157 + isoprenaline; L-NAME + L-arginine + BPC 157 + isoprenaline; isoprenaline + L-NAME + BPC 157; isoprenaline + L-NAME + L-arginine + BPC 157). Likewise, the still huge remaining pathology in the L-NAME + L-arginine animals, if further attenuated, means that the other system(s) (i.e., the BPC 157 system) may function along with the NO system (previously supposed to be immobilized by the mutual actions of combined L-NAME and L-arginine) [53]. In these terms, in the case of BPC 157 effectiveness, this means effectiveness over the background of the NO system when blocked (L-NAME), (over)stimulated (L-arginine) or immobilized (L-NAME + L-arginine) [53].

Finally, BPC 157’s beneficial effect combined with the initial reduced activity of eNOS and COX2 may provide an additional clue. Possibly, reduced activity of eNOS and COX2 may be an effect of BPC 157, in particular substituting the NO effect, whether an additive and/or synergistic effect, related to the activity of BPC 157 administration in isoprenaline-treated rats with infarction and reinfarction. Namely, eNOS, iNOS and COX2 activity are thought to be specifically up-regulated in myocardial infarction [38]. They are determined largely by underlying heart disease [38] (i.e., “positive staircase”, wherein elevated eNOS and iNOS activity is seen in cardiac muscle and NO basal release sin large arteries is diminished [38,67,68,69]).

In addition, these beneficial effects correlate with the counteraction of the isoprenaline-induced increased lipid peroxication products in the heart tissue and reduction of all of the necrosis markers commonly measured. Furthermore, BPC 157 acts as a stabilizer of cellular junctions [30] and a free radical scavenger [31,32,33]. It was shown to counteract free radical formation and lesions, in particular, in vascular occlusion studies [14,15,16,17,18,19,34,35,36,37], and normalization of the MDA level occurs in many tissues and blood in rats. Otherwise, increased levels of lipid peroxides injure blood vessels, causing increasing adherence and aggregation of platelets to the injured sites [70]. Conversely, BPC 157 maintains endothelium integrity (and thereby ascertains tissues integrity or rescues injured blood vessel functioning when confronted with vascular impairment) [14,15,16,17,18,19,34,35,36,37]. Illustratively for the counteraction of the isoprenaline myocardial infarction as well, the BPC 157 activity as a stabilizer of cellular junction (counteracting leaky gut syndrome) occurs via increasing tight junction protein ZO-1 expression and transepithelial resistance [30]. There were inhibited mRNA of inflammatory mediators (iNOS, IL-6, IFNγ and TNF-α), increased expression of HSP 70 and 90, and antioxidant proteins, such as HO-1, NQO-1, glutathione reductase, glutathione peroxidase 2 and GST-pi [30]. Thus, such a particular background strongly opposes the particular isoprenaline damaging effect, which seems to be NO system-related, as well [38]. As mentioned, in the vascular occlusion studies, BPC 157, which was always associated with strong cardioprotection and maintained heart function [14,15,16,17,18,19,20,21,22,23], exhibited a specific effect in the vessel that provides an alternative operating pathway. The expression of the Egr, Nos, Srf, Vegfr, Akt1, Plcɣ, and Kras pathways in the left ovarian vein was the key for the infrarenal occlusion-induced inferior caval vein syndrome in rats [14]. In reperfusion in stroke-induced rats [52], BPC 157 therapy counteracted both early and delayed neural hippocampal damage [52]. In hippocampal tissues, mRNA expression studies at 1 h and 24 h showed that strongly elevated (Egr1, Akt1, Kras, Src, Foxo, Srf, Vegfr2, Nos3, Nos1) and decreased (Nos2, Nfkb) gene expression (Mapk1 not activated) may be a means by which BPC 157 may act [52].

Finally, this study should emphasize the general point that animal studies, per se, must be cautious, and isoprenaline studies, in particular, must be cautious regarding their results and the relative paucity of BPC 157 clinical data [2,3,4]. On the other hand, BPC 157 was proved to be efficacious in the ulcerative colitis, both in clinical settings [71,72] as well as in experimental rat model ischemic/reperfusion vascular ulcerative colitis studies [34]. Furthermore, it has been shown to be efficacious in other ulcerative colitis models, including models induced by TNBS, cysteamine, surgery, NSAIDs [73,74,75,76,77,78,79,80] or major vessel occlusion [15,16,17,18,19,20,21,22,23]; these studies have included various species [81,82], and complications (for review see, i.e., [83]). A particular point for revealing and applying the concept of the counteraction of the isoprenaline-induced myocardial infarction in practice may be BPC 157′s beneficial effect on a larger range of arrhythmias in addition to vessel occlusion-induced arrhythmias [14,15,16,17,18,19,20,21,22,23] (i.e., digitalis [8]-, neuroleptics [9]-, succinylcholine [7]-, potassium over-dose [6]-, lithium over-dose [22]-, bupivacaine [10]-, and lidocaine [11]- induced arrhythmias). Furthermore, BPC 157 demonstrated a beneficial effect on pulmonary hypertension [12,13]; reversal of vascular occlusion goes along with cardioprotection and maintained heart function [14,15,16,17,18,19,20,21,22,23]. BPC 157 also counteracted free radical formation and lesions [14,15,16,17,18,19,34,35,36,37] and maintained NO system [53]. Finally, there are consistently effective ranges and regimens of BPC 157 (µg-ng) which are used [14,15,16,17,18,19,20,21,22,23]. These ranges and regimens may support each other’s effects, and interestingly, in the rats with the isoprenaline myocardial infarction, the same beneficial effect of the application were seen whether administration was prophylactic or curative. There is also a very safe profile (lethal dose (LD1) could be not achieved) [84], a point recently confirmed in a large study conducted by Xu and collaborators [85]. Together, these findings (for review see, i.e., [84]) may be suggestive for a physiological role (in situ hybridization and immunostaining BPC 157 in human gastrointestinal mucosa, lung bronchial epithelium, epidermal layer of the skin and kidney glomeruli) [84]. In this context, the practical indicative evidence may be even more important. BPC 157 therapy and isoprenaline, as elaborated in this study, would certainly accelerate additional studies.

## Figures and Tables

**Figure 1 biomedicines-10-00265-f001:**
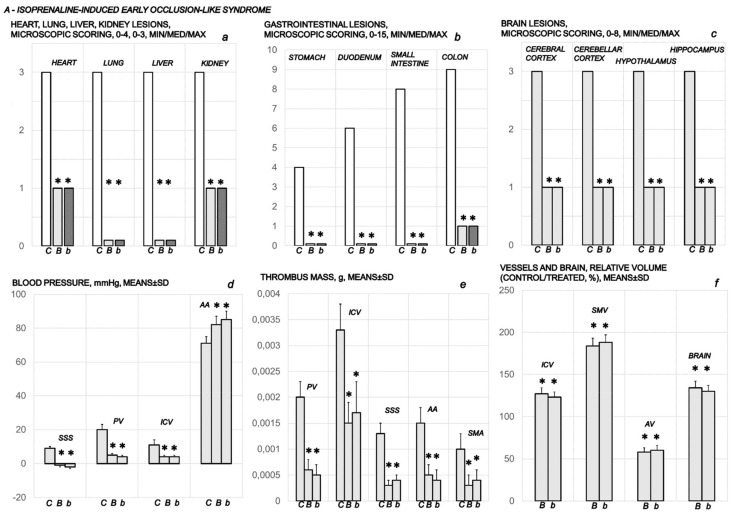
Presentation of the early occlusion-like syndrome in isoprenaline (75 mg/kg sc) treated rats at 30 min after isoprenaline administration. Medication (BPC 157 (10 ng/kg (*b*), 10 µg/kg (*B*) i.p. or saline 5.0 mL/kg (*C*) i.p.) was given 5 min after isoprenaline. Heart, lung, liver and kidney lesions microscopic scoring (**a**); gastrointestinal lesion microscopic scoring (**b**); brain lesion microscopic scoring (**c**); blood (superior sagittal sinus (SSS), portal vein (PV), inferior caval vein (ICV), abdominal aorta (AA)) pressure (**d**); thrombus mass (portal vein (PV), inferior caval vein (ICV), superior sagittal sinus (SSS), abdominal aorta (AA) and superior mesenteric artery (SMA)) (**e**); and vessels (inferior caval vein (ICV), superior mesenteric vein (SMV), azygos vein (AV) and brain relative volume (**f**) assessments were carried out. Ten rats per each experimental group. Means ± SD, min/med/max, * *p* ˂ 0.05, at least, vs. control.

**Figure 2 biomedicines-10-00265-f002:**
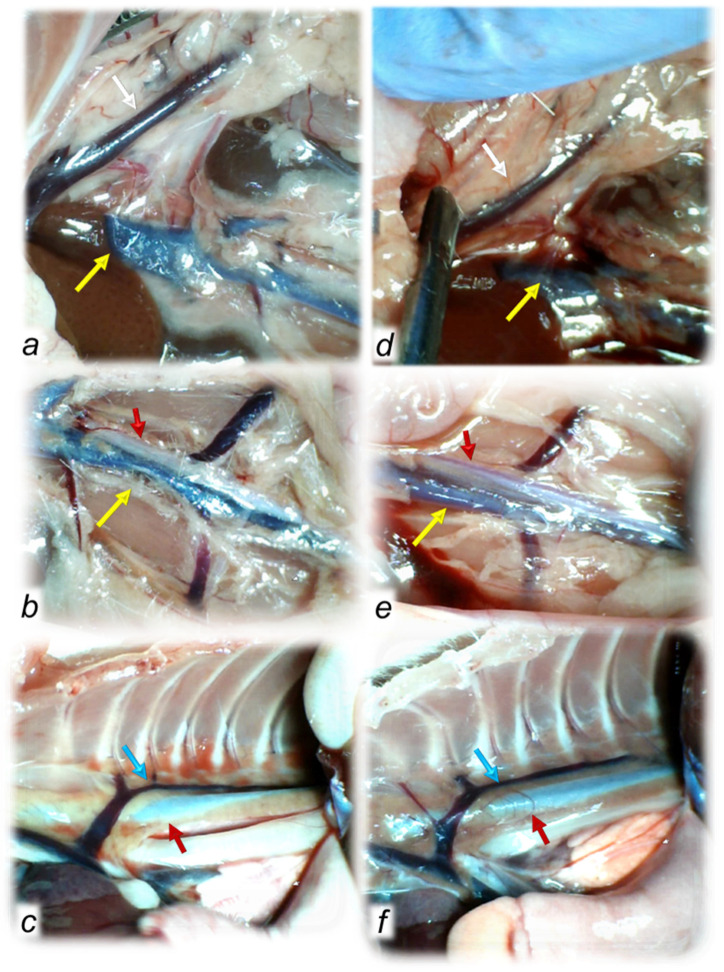
Regular timeline (30 min after isoprenaline) of vessel presentation in the isoprenaline-treated rats, disturbed (congested or collapsed) in controls (**a**–**c**) and normalized in BPC 157-treated rats (**d**–**f**), including the superior mesenteric vein (white arrows), inferior caval vein (yellow arrows), azygos vein (blue arrows) and thoracic and aorta (red arrows). Upon BPC 157 therapy, recruitment of the defensive pathways counteracted congestion of the superior mesenteric vein and inferior caval vein and counteracted failure of the azygos vein and thoracic and abdominal aorta timeline (**d**–**f**).

**Figure 3 biomedicines-10-00265-f003:**
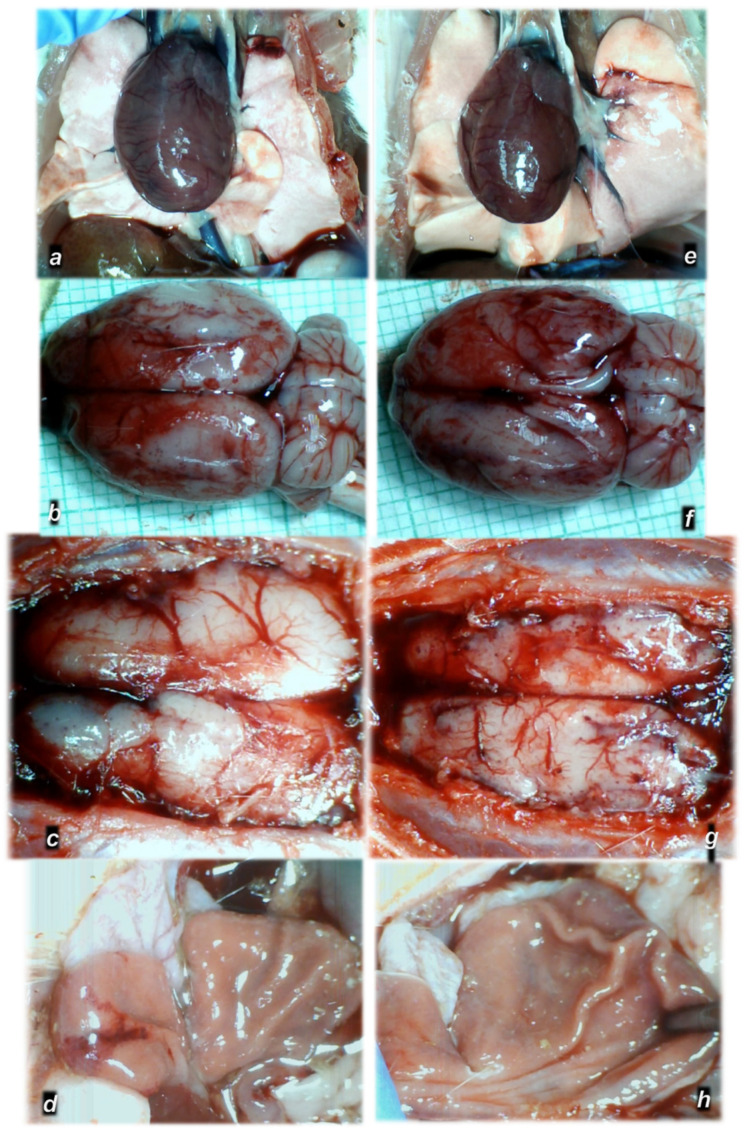
Regular timeline (30 min after isoprenaline) of organ lesion presentations in the isoprenaline-treated rats, in controls (**a**–**d**) and lesion counteraction in BPC 157-treated rats (**e**–**h**). Heart dilatation (**a**), brain swelling (**b**,**c**) and stomach lesions (**d**) in control-isoprenaline rats. Upon BPC 157 therapy, these lesions were markedly attenuated or eliminated (**e**–**h**).

**Figure 4 biomedicines-10-00265-f004:**
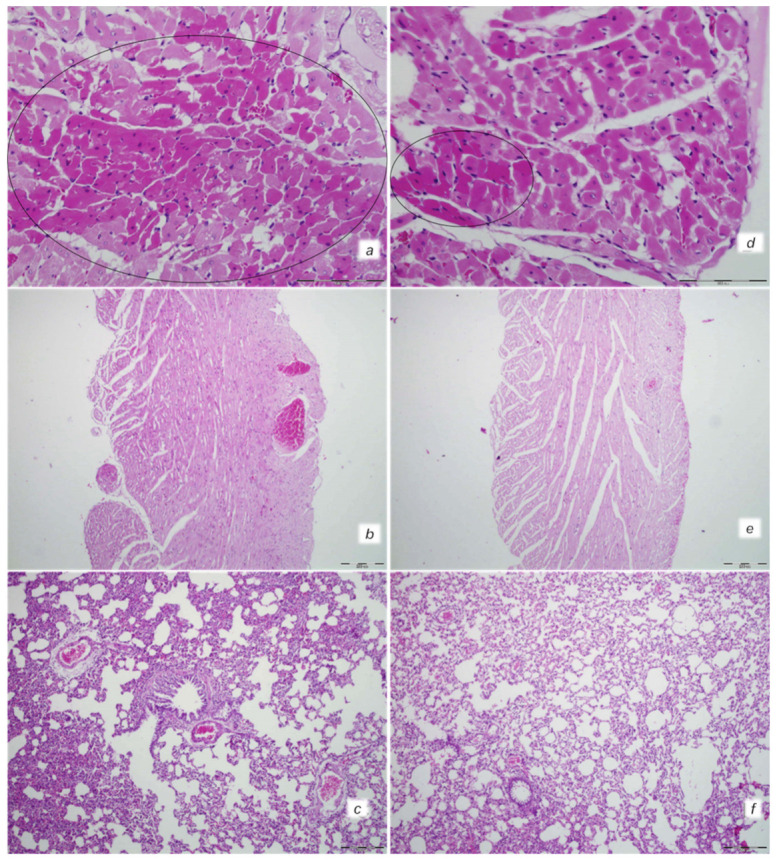
Heart and lung tissue damage. Isoprenaline 75 mg/kg at 30 min after challenge. Control rats (**a**,**b**). Myocardial congestion and confluent areas of subendocardial ischemic myocytes in the control rats (circle). Control rats (**c**). Marked congestion of the lung parenchyma in the control rats. BPC 157-treated rats (**d**,**e**). Mild myocardial congestion and only a small area of subendocardial ischemic myocytes (circle) in the BPC 157-treated rats (**f**). No changes in lung tissue in the BPC 157-treated rats. (HE staining; (**a**,**d**) magnification 400×; scale bar 100 μm; (**b**,**c**,**e**,**f**) magnification 100×; scale bar 200 μm).

**Figure 5 biomedicines-10-00265-f005:**
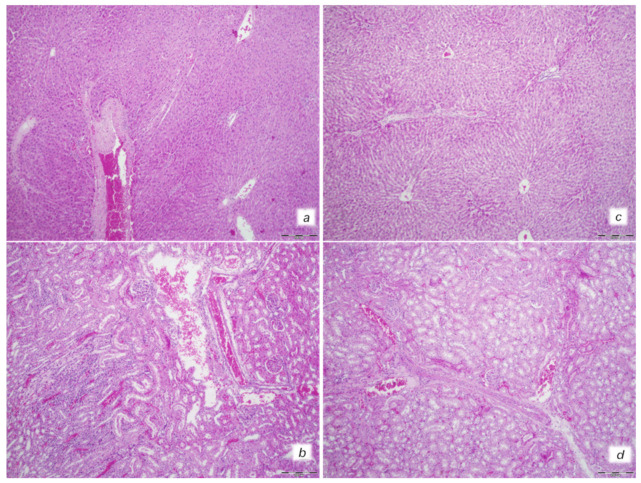
Illustrative features of liver and kidney injury. Isoprenaline 75 mg/kg at 30 min after challenge. Control rats (**a**,**b**). (**a**) Marked dilatation and congestion of blood vessels in the portal tracts as well as dilatation of central veins and sinusoids. (**b**) Conspicuous dilatation of blood vessels and congestion in the kidney tissue, as well as glomeruli. BPC 157-treated rats (**c**,**d**). (**c**) No changes in liver tissue. (**d**) Mild dilatation and congestion of blood vessels in the kidney tissue. (HE staining; magnification 100×; scale bar 200 μm).

**Figure 6 biomedicines-10-00265-f006:**
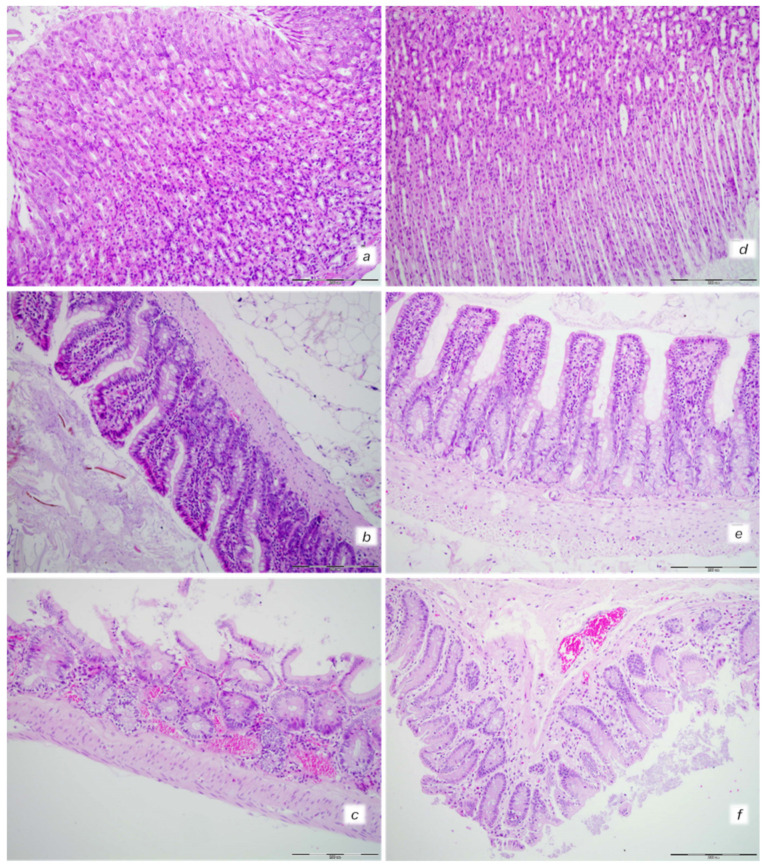
Gastrointestinal tract injury. Isoprenaline 75 mg/kg at 30 min after challenge. Control rats ((**a**) (stomach), (**b**) (small intestine), (**c**) (large intestine)). Congestion in the upper and lower digestive tract with a tendency pronounced in the lower digestive tract. BPC 157-treated rats ((**d**) (stomach), (**e**) (small intestine), (**f**) (large intestine)). No changes in the stomach and small intestine with mild congestion of the colon. (HE staining; magnification 200×; scale bar 200 μm).

**Figure 7 biomedicines-10-00265-f007:**
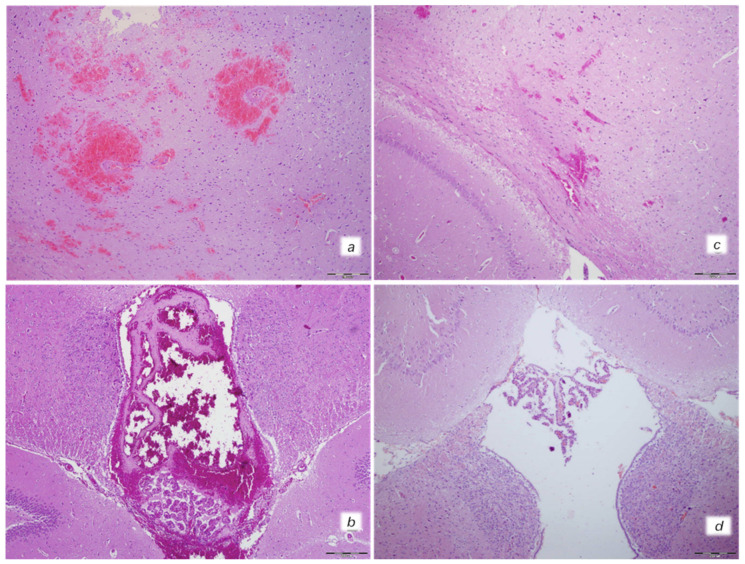
Neuropathological changes of cerebral area. Isoprenaline 75 mg/kg at 30 min after challenge. Control group (**a**,**b**). (**a**) Large areas of intracerebral hemorrhage. B. Intraventricular hemorrage in the 3rd ventricle. BPC 157 treated rats (**c**,**d**). (**c**) Minor intracerebral hemorrhage. (**d**) No hemorrhage in the 3rd ventricle. (HE; magnification 100×; scale bar 200 μm).

**Figure 8 biomedicines-10-00265-f008:**
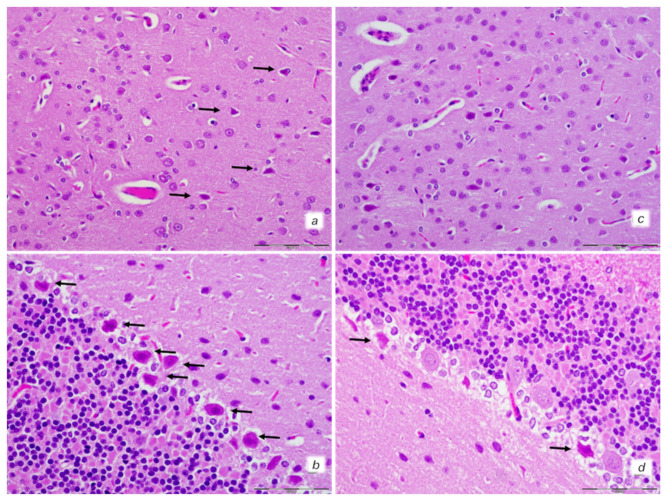
Neuropathological features of cerebral and celebellar cortex. Isoprenaline 75 mg/kg at 30 min after challenge. Control rats (**a**,**b**). (**a**) Some karyopyknotic cells in cerebral cortex (black arrows). (**b**) Moderate karyopyknosis and degeneration of Purkinje cells of the cerebellar cortex. BPC 157-treated rats (**c**,**d**). (**c**) No karyopyknotic cells in cerebral cortex. (**d**) Only a few karyopyknotic and degenerated Purkinje cells of the cerebellar cortex were found. (HE; magnification 400×; scale bar 50 μm).

**Figure 9 biomedicines-10-00265-f009:**
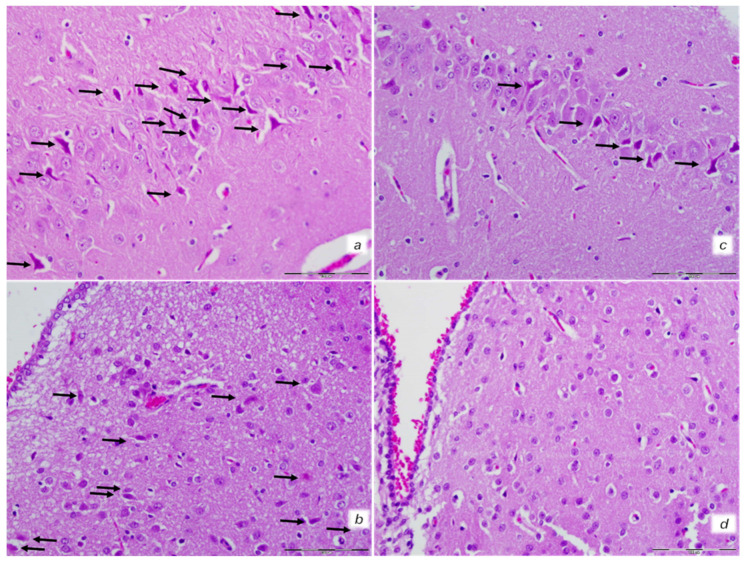
Neuropathological features of the hippocampus and hypothalamus. Isoprenaline 75 mg/kg at 30 min after challenge. Control rats (**a**,**b**). (**a**) Marked karyopyknosis of pyramidal cells of the hippocampus (black arrows). (**b**) Rare karyopyknotic cells in hypothalamic area. BPC 157-treated rats (**c**,**d**). (**c**) Small number karyopyknosis of pyramidal cells of the hippocampus. (**d**) No changes in hipothalamic area. (HE staining; magnification 400×; scale bar 100 μm).

**Figure 10 biomedicines-10-00265-f010:**
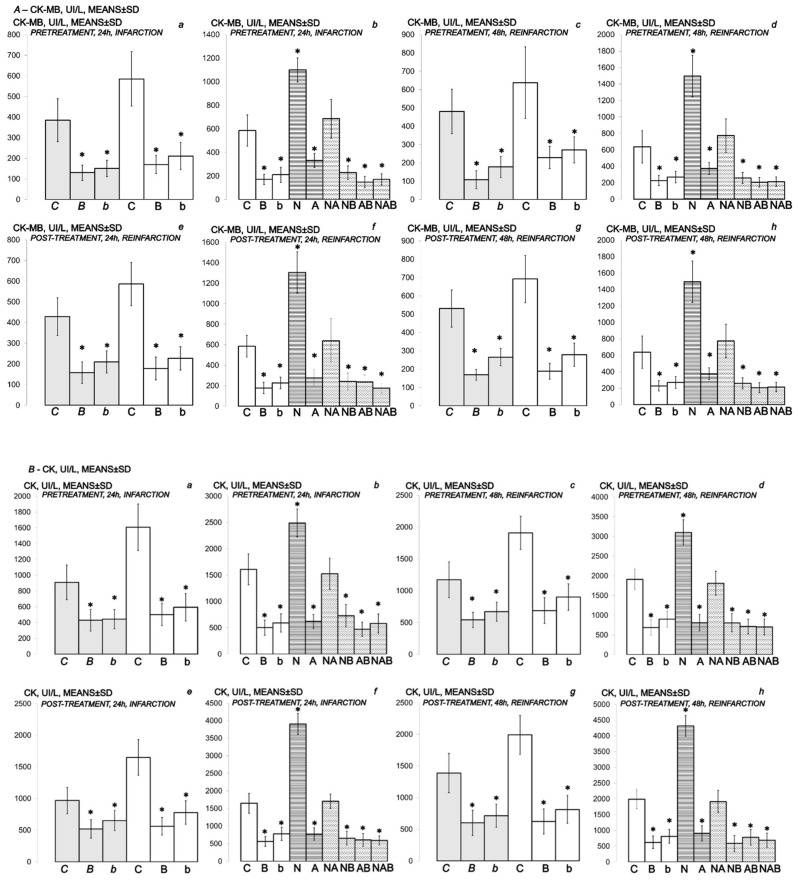
(**A**,**B**) Myocardial infarction (one isoprenaline challenge) and re-infarction (two isoprenaline subsequent challenges) in isoprenaline-treated rats (75 mg/kg sc (gray bars), 150 mg/kg sc (white bars)). Serum values of cardioselective enzymes *A-CK-MB* (UI/L) and *B-CK* (UI/L). Rats treated with the smaller dose of isoprenaline, 75 mg/kg sc (gray bars, italic letters), received medication (BPC 157 (10 ng/kg (*b*), 10 µg/kg (*B*) i.p. or saline (5 mL/kg i.p. (*C*)). Rats treated with the higher dose of isoprenalines, 150 mg/kg sc (white bars, normal letters), received medication (BPC 157 (10 ng/kg (b), 10 µg/kg (B) i.p. or saline (5 mL/kg i.p. (C)) or L-NAME (5 mg/kg i.p. (N)), L-arginine (200 mg/kg i.p. (A)) (horizontal line bars) or these agents in combination (L-NAME + L-arginine (NA), L-NAME + BPC 157 (NB), L-arginine + BPC 157 (AB), L-NAME + L-arginine + BPC 157 (NAB)) (dashed horizontal lines bars). Therapy was given (i) 30 min before isoprenaline (*PRETREATMENT*), prophylactic regimen (**a**–**d**), or, alternatively, (ii) at 5 min after isoprenaline (75 mg/kg s.c. or 150 mg/kg s.c.), at day 1 and at day 2 (*POST-TREATMENT*, therapeutic regimen (**e**–**h**)). Ten rats per each experimental group. * *p* < 0.05 vs. control, at least.

**Figure 11 biomedicines-10-00265-f011:**
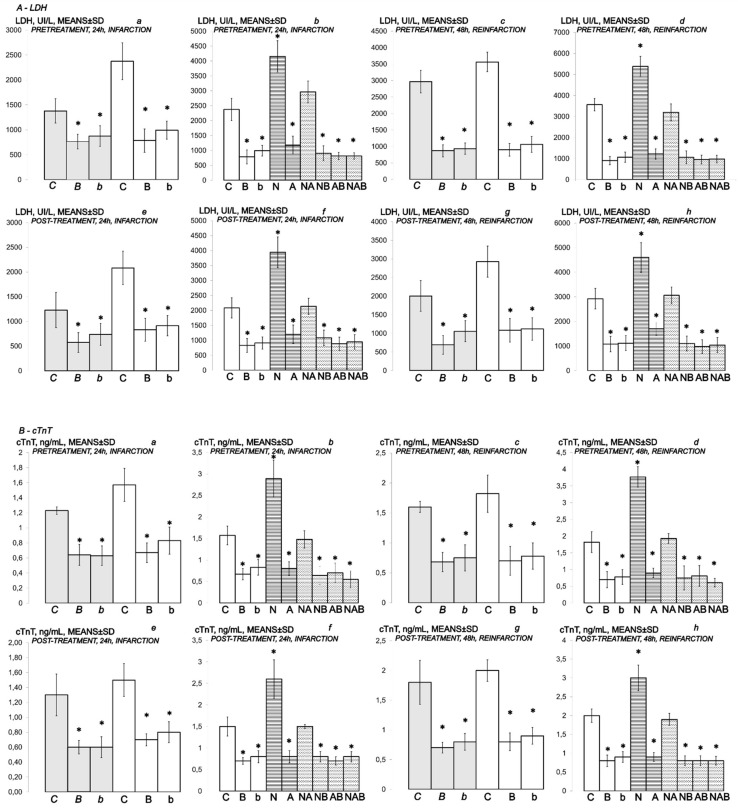
(**A**,**B**) Myocardial infarction (one isoprenaline challenge) and re-infarction (two isoprenaline subsequent challenges) in isoprenaline- treated rats (75 mg/kg sc (gray bars), 150 mg/kg sc (white bars)). Serum values of cardioselective enzymes *A-LDH* (UI/L) and *B-cTnT* ng/mL. Rats treated with the smaller dose of isoprenaline, 75 mg/kg sc (gray bars, italic letters), received medication (BPC 157 (10 ng/kg (*b*), 10 µg/kg (*B*) i.p. or saline (5 mL/kg i.p. (*C*)). Rats treated with the higher dose of isoprenaline, 150 mg/kg sc (white bars, normal letters), received medication (BPC 157 (10 ng/kg (b), 10 µg/kg (B) i.p. or saline (5 mL/kg i.p. (C)) or L-NAME (5 mg/kg i.p. (N)), L-arginine (200 mg/kg i.p. (A)) (horizontal line bars) or these agents combination (L-NAME + L-arginine (NA), L-NAME + BPC 157 (NB), L-arginine + BPC 157 (AB), L-NAME + L-arginine + BPC 157 (NAB)) (dashed horizontal lines bars). Therapy was given (i) 30 min before isoprenaline (*PRETREATMENT*, prophylactic regimen (**a**–**d**)), or, alternatively, (ii) at 5 min after isoprenaline (75 mg/kg s.c. or 150 mg/kg s.c.), at day 1 and at day 2 (*POST-TREATMENT*, therapeutic regimen (**e**–**h**)). Ten rats per each experimental group. * *p* < 0.05 vs. control, at least.

**Figure 12 biomedicines-10-00265-f012:**
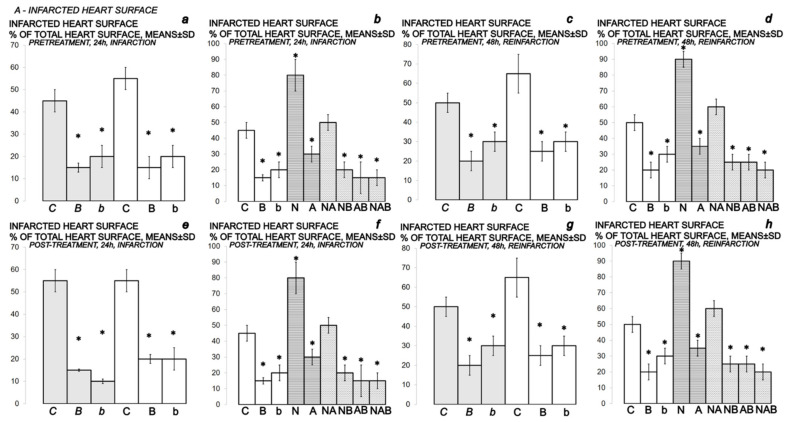
Myocardial infarction (one isoprenaline challenge) and re-infarction (two isoprenaline subsequent challenges) in isoprenaline-rats (75 mg/kg sc (gray bars), 150 mg/kg sc (white bars)), gross heart lesions presentation, *A- INFARCTED HEART SURFACE*. Isoprenaline-rats treated with the smaller dose, 75 mg/kg sc (gray bars, italic letters), received medication (BPC 157 (10 ng/kg (*b*), 10 µg/kg (*B*) i.p. or saline (5 mL/kg i.p. (*C*)). Isoprenaline-rats treated with the higher dose, 150 mg/kg sc (white bars, normal letters), received medication (BPC 157 (10 ng/kg (b), 10 µg/kg (B) i.p. or saline (5 mL/kg i.p. (C)) or L-NAME (5 mg/kg i.p. (N)), L-arginine (200 mg/kg i.p. (A)) (horizontal line bars) or these agents in combination (L-NAME + L-arginine (NA), L-NAME + BPC 157 (NB), L-arginine + BPC 157 (AB), L-NAME + L-arginine + BPC 157 (NAB)) (dashed horizontal lines bars). Therapy was given (i) 30 min before isoprenaline (*PRETREATMENT*, prophylactic regimen (**a**–**d**)), or, alternatively, (ii) at 5 min after isoprenaline (75 mg/kg s.c. or 150 mg/kg s.c.), at day 1 and at day 2 (*POST-TREATMENT*, therapeutic regimen (**e**–**h**)). Ten rats per each experimental group. * *p* < 0.05 vs. control, at least.

**Figure 13 biomedicines-10-00265-f013:**
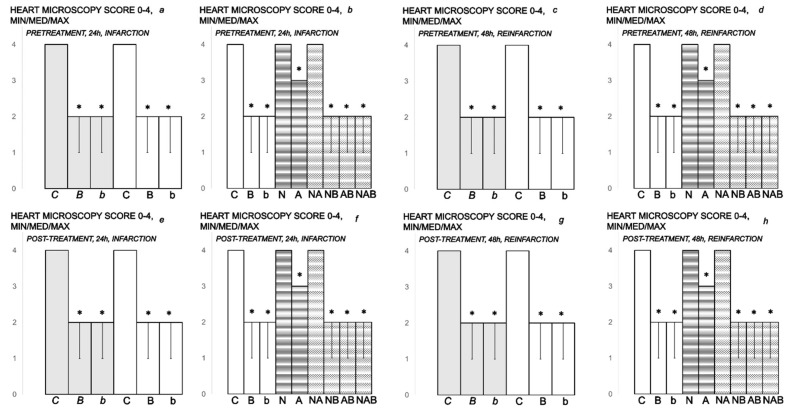
Myocardial infarction (one isoprenaline challenge) and re-infarction (two isoprenaline subsequent challenges) in isoprenaline-treated rats (75 mg/kg sc (gray bars), 150 mg/kg sc (white bars)) and heart lesion microscopy presentation, *A- HEART MICROSCOPY*. Rats treated with the smaller dose of isoprenaline, 75 mg/kg sc (gray bars, italic letters), received medication (BPC 157 (10 ng/kg (*b*), 10 µg/kg (*B*) i.p. or saline (5 mL/kg i.p. (*C*)). Rats treated with the higher dose of isoprenaline, 150 mg/kg sc (white bars, normal letters), received medication (BPC 157 (10 ng/kg (b), 10 µg/kg (B) i.p. or saline (5 mL/kg i.p. (C)) or L-NAME (5 mg/kg i.p. (N)), L-arginine (200 mg/kg i.p. (A)) (horizontal line bars) or these agents in combination (L-NAME + L-arginine (NA), L-NAME + BPC 157 (NB), L-arginine + BPC 157 (AB), L-NAME + L-arginine + BPC 157 (NAB)) (dashed horizontal lines bars). Therapy was given (i) 30 min before isoprenaline (*PRETREATMENT*, prophylactic regimen (**a**–**d**)), or, alternatively, (ii) at 5 min after isoprenaline (75 mg/kg s.c. or 150 mg/kg s.c.), at day 1 and at day 2 (*POST-TREATMENT*, therapeutic regimen (**e**–**h**)). Ten rats per each experimental group. * *p* < 0.05 vs. control, at least.

**Figure 14 biomedicines-10-00265-f014:**
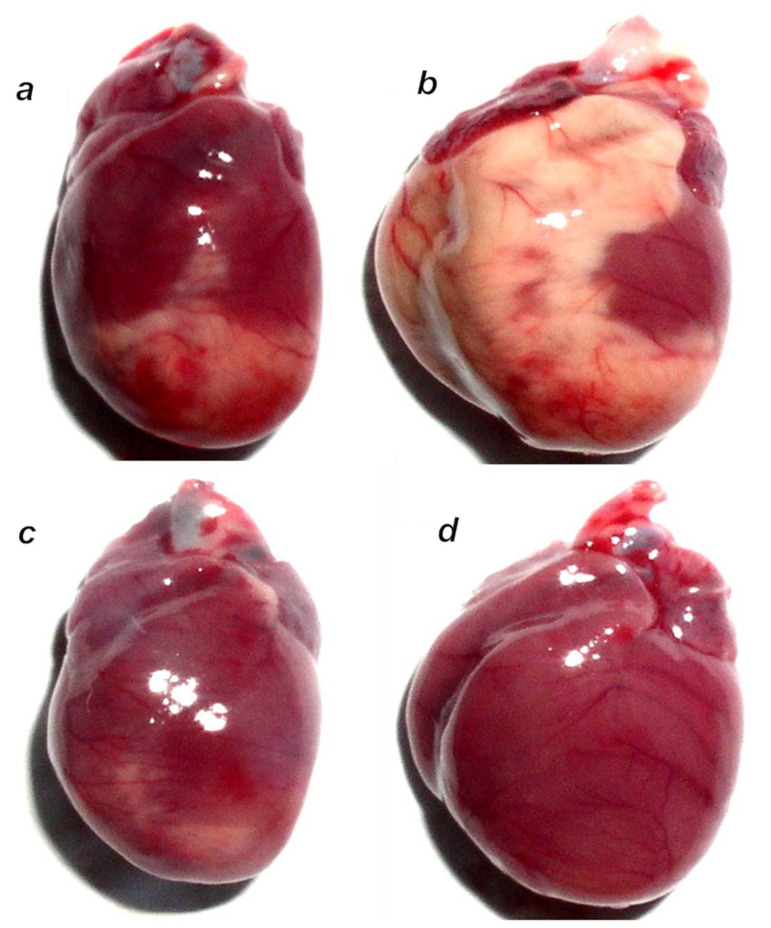
Illustrative gross presentation of anterior heart surface in rats at 24 h after isoprenaline. Rats challenged once with isoprenaline 150 mg/kg s.c., mediation at 30 min before isoprenaline. (**a**) Significant myocardial infarction in controls that received saline (5 mL/kg i.p.). (**b**) L-NAME (5 mg/kg i.p.)-pretreated rats presented with large infarcted area extended to the anterior, lateral, apical and inferior heart surfaces. (**c**) L-arginine (200 mg/kg i.p.)-pretreated rats presented with a lesser infracted area. (**d**) BPC 157 (10 μg/kg i.p.)-pretreated rats presented with no visible infracted area.

**Figure 15 biomedicines-10-00265-f015:**
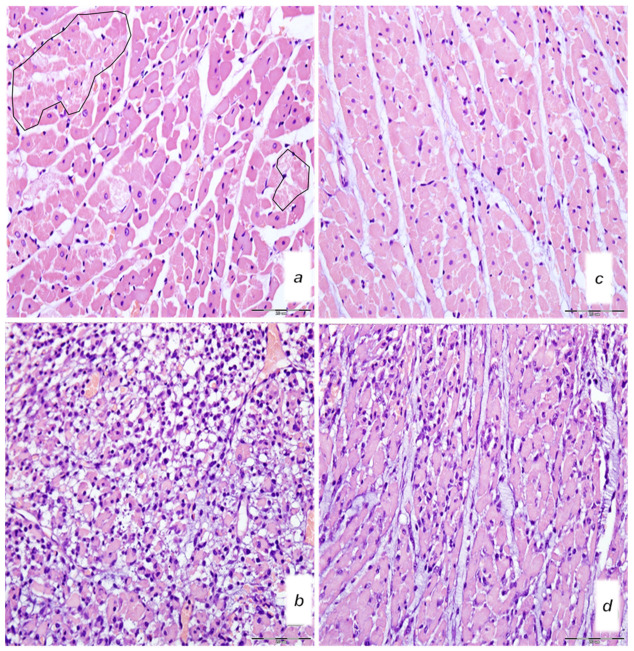
Microscopy presentation of heart injury. Rats challenged once with isoprenaline 150 mg/kg s.c., mediation at 30 min before isoprenaline (**a–d**). Upper 24 h. (**a**,**c**) Single and small groups of necrotic myocites without inflammation (irregular circles) (magnification 400×, scale bar 20 μm) (saline (5 mL/kg i.p, left)) (**a**); single and small groups of necrotic myocites without inflammation (magnification 400x, scale bar 20 μm) (BPC 157 (10 μg/kg i.p., right)) (**c**). Low (**b**,**d**). Pronounced inflammatory infiltrate with necrotic myocites (magnification 400×, scale bar 20 μm) (saline (5 mL/kg i.p, left)) (**b**). Moderate inflammatory infiltrate affecting up to half of the septal thickness (magnification 400×, scale bar 20 μm) (BPC 157 (10 μg/kg i.p., right)). (**d**). (HE staining).

**Figure 16 biomedicines-10-00265-f016:**
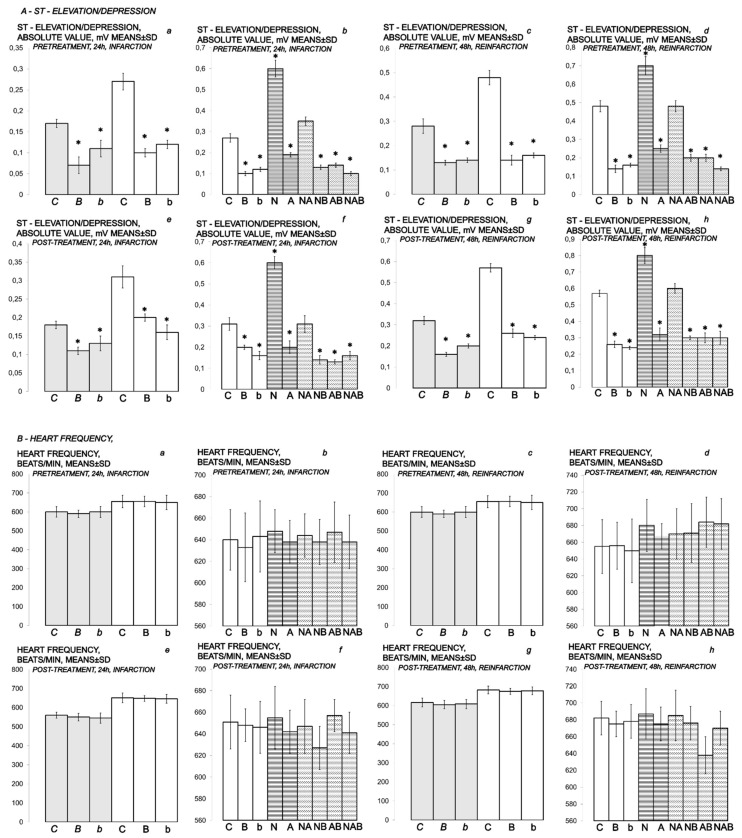
(**A**,**B**) Myocardial infarction (one isoprenaline challenge) and re-infarction (two isoprenaline subsequent challenges) in isoprenaline-treated rats (75 mg/kg sc (gray bars), 150 mg/kg sc (white bars)). Gross heart lesions presentation; *A*–*ST-ELEVATION/DEPRESSION* (mV), *B*–*HEART FREQUENCY* (beats/min). Rats treated with the smaller dose of isoprenaline, 75 mg/kg sc (gray bars, italic letters), received medication (BPC 157 (10 ng/kg (*b*), 10 µg/kg (*B*) i.p. or saline (5 mL/kg i.p. (*C*)). Rats treated with the higher dose of isoprenaline, 150 mg/kg sc (white bars, normal letters), received medication (BPC 157 (10 ng/kg (b), 10 µg/kg (B) i.p. or saline (5 mL/kg i.p. (C)) or L-NAME (5 mg/kg i.p. (N)), L-arginine (200 mg/kg i.p. (A)) (horizontal line bars) or these agents in combination (L-NAME + L-arginine (NA), L-NAME + BPC 157 (NB), L-arginine + BPC 157 (AB), L-NAME + L-arginine + BPC 157 (NAB)) (dashed horizontal lines bars). Therapy was given (i) 30 min before isoprenaline (*PRETREATMENT*, prophylactic regimen (**a**–**d**)), or, alternatively, (ii) at 5 min after isoprenaline (75 mg/kg s.c. or 150 mg/kg s.c.), at day 1 and at day 2 (*POST-TREATMENT*, therapeutic regimen (**e**–**h**)). Ten rats per each experimental group. * *p* < 0.05 vs. control, at least.

**Figure 17 biomedicines-10-00265-f017:**
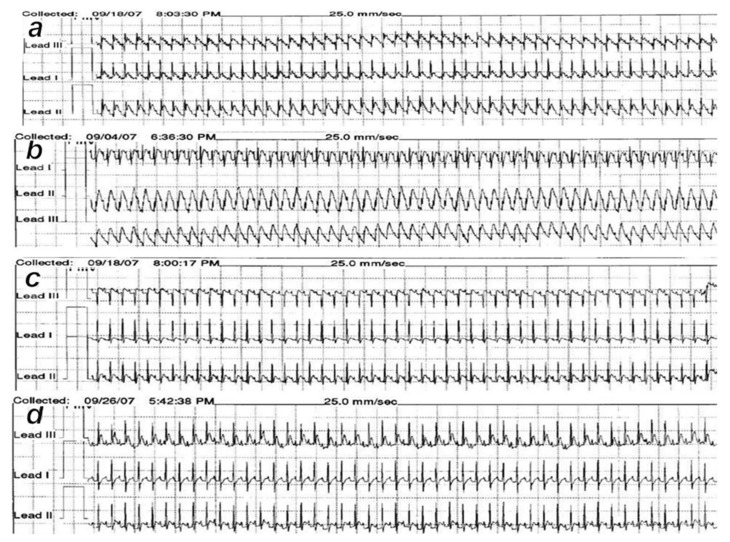
ECG presentation. Rats challenged once with isoprenaline 150 mg/kg s.c., medication at 30 min before isoprenaline. (**a**) ST elevation in all 3 standard leads in controls that received saline (5 mL/kg ip) 24 h after first isoprenaline challenge. (**b**) L-NAME (5 mg/kg i.p.) medication at 30 min before isoprenaline presented with ST elevation in all 3 standard leads. (**c**) L-arginine (200 mg/kg i.p.) presented with less expressed ischemic ECG changes at 24 h after initial isoprenaline dose. (**d**) BPC 157 (10 µg/kg i.p.) presented with no ST-T ischemic changes.

**Figure 18 biomedicines-10-00265-f018:**
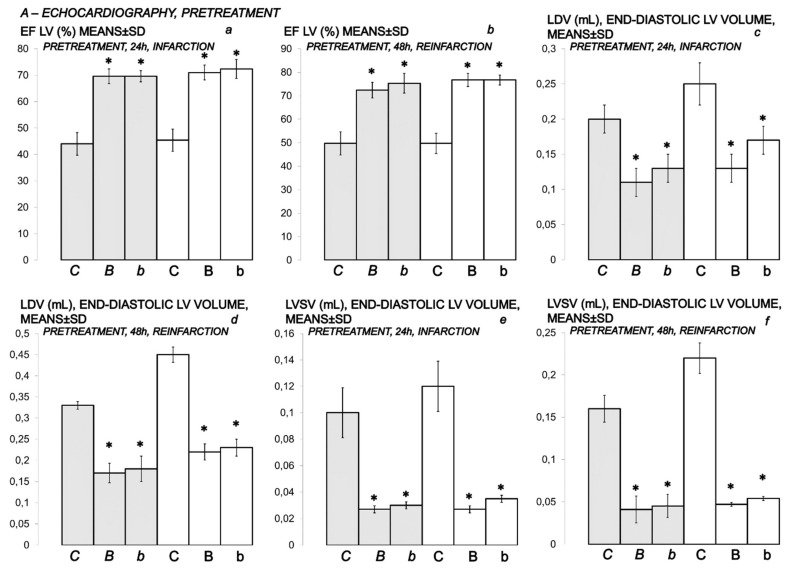
(**A**,**B**) Myocardial infarction (1 isoprenaline challenge) and re-infarction (2 isoprenaline subsequent challenges) in isoprenaline-rats (75 mg/kg sc (gray bars), 150 mg/kg sc (white bars)). Gross heart lesions presentation. A–ECHOCARDIOGRAPHY, PRETREATMENT; B–ECHOCARDIOGRAPHY, POST-TREATMENT. Isoprenaline-rats treated with the smaller dose, 75 mg/kg sc (gray bars, italic letters), received medication (BPC 157 (10 ng/kg (*b*), 10 µg/kg (*B*) i.p. or saline (5 mL/kg i.p. (*C*)). Isoprenaline-rats treated with the higher dose, 150 mg/kg sc (white bars, normal letters), received medication (BPC 157 (10 ng/kg (b), 10 µg/kg (B) i.p. or saline (5 mL/kg i.p. (C)). Therapy was given (i) 30 min before isoprenaline (PRETREATMENT, prophylactic regimen (EF LV (%), 24 h, infarction (**a**), 48 h reinfarction (**b**), LDV (mL) diastolic volume, 24 h, infarction (**c**), 48 h reinfarction (**d**), LVSV (mL), end-diastolic volume, 24 h, infarction (**e**), 48 h reinfarction (**f**)) or, alternatively, (ii) at 5 min after isoprenaline (75 mg/kg s.c. or 150 mg/kg s.c.), at day 1 and at day 2 (POST-TREATMENT, therapeutic regimen (EF LV (%), 24 h, infarction (**a**), 48 h reinfarction (**b**), LDV (mL) diastolic volume, 24 h, infarction (**c**), 48 h reinfarction (**d**), LVSV (mL), end-diastolic volume, 24 h, infarction (**e**), 48 h reinfarction (**f**)). Ten rats per each experimental group. * *p* < 0.05 vs. control, at least.

**Figure 19 biomedicines-10-00265-f019:**
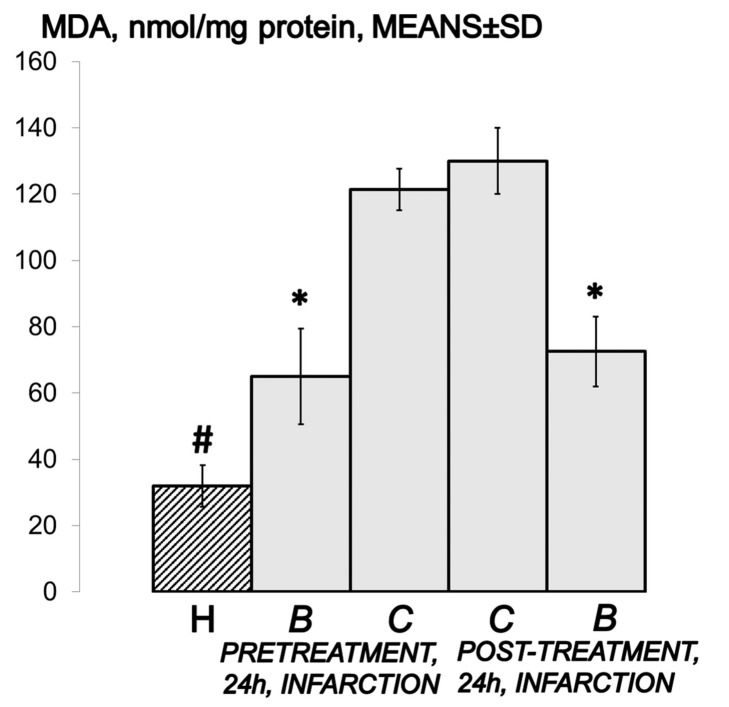
MDA levels in heart tissue. MDA levels in the heart tissue of isoprenaline-treated rats at 24 h after challenge (isoprenaline 75 mg/kg sc) (gray bars), determined by quantifying thiobarbituric acid (TBA) reactivity as malondialdehyde (MDA) equivalents. BPC 157 (10 µg/kg ip) (*B*) or an equal volume of saline (5 mL/kg ip) (controls) (*C*) was applied to the isoprenaline-treated rats before or after the challenge. Ten rats per each experimental group. * *p* < 0.05 vs. saline; # *p* < 0.05 healthy (dashed bar) vs. isoprenaline-treated rats (H).

**Figure 20 biomedicines-10-00265-f020:**
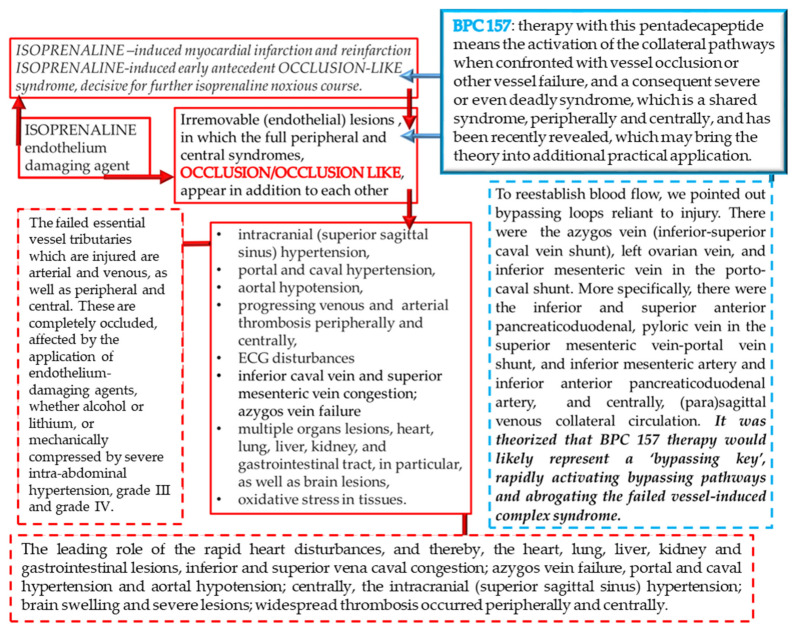
Summary of the background of the isoprenaline effects (red frameworks) and therapy potential of BPC 157 application (blue frameworks) [14,15,16,17,18,19,20,21,22,23]. It is likely that isoprenaline, via subcutaneous administration, indicates irremovable (endothelial) lesions (red frameworks), in which the full peripheral and central syndromes appear in addition to each other (red frameworks) as also noted with intragastric absolute alcohol-induced, over-dose lithium-induced and maintained intra-abdominal hypertension-induced occlusion-like syndrome (red dashes frameworks) [21,22,23]. Due to the prime peripheral or central lesion, these lesions should appear as an essential cause–consequence cycle (red dashes frameworks) [14,15,16,17,18,19,20,21,22,23]. BPC 157 therapy means the activation of the collateral pathways when confronted with vessel occlusion or other vessel failure, and a consequent severe or even deadly syndrome, which is a shared syndrome, peripherally and centrally, and has been recently revealed, which may bring the theory into additional practical application (blue framework). Using the BPC 157 regimen, we commonly observed the counteraction of intracranial (superior sagittal sinus) hypertension, portal and caval hypertension, aortal hypotension, and progressing venous and arterial thrombosis peripherally and centrally; furthermore, ECG disturbances were attenuated. Multiple organ lesions, were also attenuated, including those of the heart, lung, liver, kidney, and gastrointestinal tract, in particular, as well as brain lesions. Oxidative stress in tissues were also attenuated (blue dashes framework). Thus, this may be suitable background for the counteraction of the early isoprenaline-induced occlusion-like syndrome, and thereby, isoprenaline-induced myocardial infarction and reinfarction.

**Table 1 biomedicines-10-00265-t001:** GAPDH, iNOS, eNOS and COX-2 primers.

Gene	Nucleotide Sequence	Product Size	GenBank Accession No.
GAPDH Glyceraldehyde-3-phosphate dehydrogenase	Sense: TGGCAAGTTCAACGGCACAGT Antisense: TTTGGCCTCACCCTTCAGGT	193 bp	XM_221353
iNOS (NOS-2) Inducible nitric oxide synthase	Sense: TTGGAGCGAGTTGTGGATTGTTGTTC Antisense: GGTGAGGGCTTGCCTGAGTGAGC	126 bp	NM_012611
eNOS (NOS-3) Endothelial nitric oxide synthase	Sense: CTGGCAAGACCGATTACACGA Antisense: TCAGGAGGTCTTGCACATAGG	206 bp	NM_021838
COX-2 Cyclooxygenase-2	Sense: CTGTATCCCGCCCTGCTGGTG Antisense: CCACTTCTCCTCCGAAGGTGC	157 bp	AF233596

**Table 2 biomedicines-10-00265-t002:** Clinical presentation, respiratory frequency/min (means ± SD); peripheral edema (scored 0–3, min/med/max) in isoprenaline-treated rats treated with BPC 157, L-arginine, and L-NAME, given alone or together. Prophylactic regimen. Therapeutic regimen. Isoprenaline (75 mg/kg or 150 mg/kg s.c., given once (at time 0) or twice, at 0 and 24 h). First challenge for initial infarction induction, assessed at 24 h thereafter; then, at 24 h, the next application (isoprenaline given twice, at 0 and 24 h) for reinfarction induction (assessment at the end of the subsequent 24 h period). Medication (/kg, i.p.) (BPC 157 (10 µg, 10 ng), L-NAME (5 mg), L-arginine (200 mg) (given alone or together) or saline (5 mL) (control)) at (i) 30 min before; or, alternatively (ii) at 5 min after isoprenaline. * Means ± S.D. vs. control, at least *p* < 0.05, 10 rats per each experimental group.

Prophylactic Regimen					
Medication /kg i.p.	Noxious procedure mg/kg s.c.	Clinical status in time after initial isoprenaline application			
Prophylactic regimen	Isoprenaline at “0” time and “24 h” time	Respiratory frequency/min, means ± SD		Peripheral edema, scored 0–3, min/med/max	
Medication at 30 min before isoprenaline		24 h (one isoprenaline challenge) infarction	48 h (two isoprenaline challenges) reinfarction	24 h (one isoprenaline challenge) infarction	48 h (two isoprenaline challenges) reinfarction
Saline 5 mL	Isoprenaline 75	138 ± 41	158 ± 47	1/2/3	2/3/3
BPC 157 10 µg		77 ± 23 *	81 ± 24 *	0/0/0 *	0/1/2 *
BPC 157 10 ng		80 ± 24 *	90 ± 27 *	0/0/1 *	1/1/2 *
Saline 5 mL	Isoprenaline 150	144 ± 43	133 ± 49	1/2/3	2/3/3
BPC 157 10 µg		83 ± 25 *	93 ± 28 *	0/0/0 *	0/1/2 *
BPC 157 10 ng		95 ± 28 *	99 ± 29 *	0/0/1 *	1/1/2 *
L-NAME 5 mg		161 ± 48 *	177 ± 53 *	2/3/3 *	2/3/3 *
L-arginine 200 mg		92 ± 27 *	95 ± 28 *	0/1/2 *	1/1/2 *
L-NAME 5 mg + L-arginine 200 mg		142 ± 42	159 ± 47	1/2/3	2/3/3
BPC 157 10 µg + L-NAME 5 mg		89 ± 26 *	94 ± 28 *	0/0/0 *	0/1/1 *
BPC 157 10 µg + L-arginine 200 mg		81 ± 24 *	92 ± 27 *	0/0/0 *	0/1/2 *
BPC 157 10 µg + L-NAME 5 mg + L-arginine 200 mg		85 ± 25 *	97 ± 29 *	0/0/1 *	1/1/2 *
**Therapeutic regimen**					
Noxious procedure mg/kg s.c.	Medication /kg i.p.	Clinical status in time after initial isoprenaline application			
Isoprenaline at “0” time and “24 h” time	Therapeutic regimen	Respiratory frequency/min, means ± SD		Peripheral edema, scored 0–3, min/med/max	
	Medication at 5 min after isoprenaline	24 h (one isoprenaline challenge) infarction	48 h (two isoprenaline challenges) reinfarction	24 h (one isoprenaline challenge) infarction	48 h (two isoprenaline challenges) reinfarction
Isoprenaline 75	Saline 5 mL	144 ± 43	168 ± 50	1/2/3	2/3/3
	BPC 157 10 µg	81 ± 24 *	90 ± 27 *	0/0/1 *	0/1/1 *
	BPC 157 10 ng	85 ± 25 *	96 ± 28 *	0/0/1 *	0/1/2 *
Isoprenaline 150	Saline 5 mL	155 ± 46	170 ± 51	2/2/3	2/3/3
	BPC 157 10 µg	92 ± 27 *	100 ± 30 *	0/0/1 *	0/1/1 *
	BPC 157 10 ng	100 ± 30 *	105 ± 28 *	0/0/1 *	1/1/2 *
	L-NAME 5 mg	176 ± 52 *	192 ± 57 *	2/3/3 *	2/3/3 *
	L-arginine 200 mg	90 ± 27 *	100 ± 30 *	0/1/2 *	1/1/2 *
	L-NAME 5 mg + L-arginine 200 mg	152 ± 45	168 ± 51	1/2/2	2/3/3
	BPC 157 10 µg + L-NAME 5 mg	96 ± 28 *	105 ± 31 *	0/0/1 *	1/1/2 *
	BPC 157 10 µg + L-arginine 200 mg	98 ± 29 *	103 ± 32 *	0/0/1 *	1/1/2 *
	BPC 157 10 µg + L-NAME 5 mg + L-arginine 200 mg	90 ± 27 *	101 ± 35 *	0/0/1 *	1/1/2 *

**Table 3 biomedicines-10-00265-t003:** Mortlity rate (expressed as percentage of total number that survived given regimens (20 rats per initial group)) in isoprenaline-treated rats treated with BPC 157, L-arginine, and L-NAME, given alone or together. Prophylactic regimen. Therapeutic regimen. Isoprenaline (75 mg/kg or 150 mg/kg s.c., given once (at time 0) or twice, at 0 and 24 h). First challenge for initial infarction induction, assessed at 24 h thereafter; and then at the point of 24 h, the next application (isoprenaline given twice, at 0 and 24 h) for reinfarction induction (assessment at the end of the subsequent 24 h period). Medication (/kg, i.p.) (BPC 157 (10 µg, 10 ng), L-NAME (5 mg), L-arginine (200 mg) (given alone or together) or saline (5 mL) (control)) at (i) 30 min before; or, alternatively (ii) at 5 min after isoprenaline. vs. control, * at least *p* < 0.05, 10 rats per each experimental group.

Medication and Noxious Procedure		Mortality Rate Expressed as Percentage of Total Number that Survived Given Regimens (20 Rats per Initial group) Prophylactic Regimen		Medication /kg i.p. at 5 min after Isoprenaline	Mortality Rate Expressed as Percentage of Total Number that Survived Given Regimens (20 Rats per Initial Group) Therapeutic Regimen	
Medication /kg i.p. at 30 min before Isoprenaline	Isoprenaline Challenge /kg s.c. (time “0”) (time “24 h”)	Time after initiation of the isoprenaline noxious procedure			Time after Initiation of the Isoprenaline Noxious Procedure	
		24 h (One Isoprenaline Challenge) Infarction	48 h (two Isoprenaline Challenges) Reinfarction		24 h (One Isoprenaline Challenge) Infarction	48 h (Two Isoprenaline Challenges) Reinfarction
Saline 5 mL	Isoprenaline 75 mg	40	50	Saline 5 mL	45	55
BPC 157 10 µg		0 *	0 *	BPC 157 10 µg	0 *	0 *
BPC 157 10 ng		0 *	0 *	BPC 157 10 ng	0 *	0 *
Saline 5 mL	Isoprenaline 150 mg	45	63	Saline 5 mL	50	60
BPC 157 10 µg		0 *	0 *	BPC 157 10 µg	0 *	0 *
BPC 157 10 ng		0 *	0 *	BPC 157 10 ng	0 *	0 *
L-NAME 5 mg		75 *	80 *	L-NAME 5 mg	75 *	80 *
L-arginine 200 mg		5 *	5.3 *	L-arginine 200 mg	10 *	11.1 *
L-NAME 5 mg + L-arginine 200 mg		50	60	L-NAME 5 mg + L-arginine 200 mg	50	60
BPC 157 10 µg + L-NAME 5 mg		0 *	0 *	BPC 157 10 µg + L-NAME 5 mg	0 *	0 *
BPC 157 10 µg + L-arginine 200 mg		0 *	0 *	BPC 157 10 µg + L-arginine 200 mg	0 *	0 *
BPC 157 10 µg/kg + L-NAME 5 mg+ L-arginine 200 mg		0 *	0 *	BPC 157 10 µg/kg + L-NAME 5 mg + L-arginine 200 mg	0 *	0 *

**Table 4 biomedicines-10-00265-t004:** eNOS/GAPDH, iNOS/GAPDH, COX-2/GAPDH. Densitometric analysis of RT-PCR products showing the effect of BPC 157 on eNOS and COX-2 mRNA levels in heart septum tissue. Results were expressed as the ratio of the optical density of the eNOS PCR and COX-2 products to the density of the corresponding GAPDH PCR products and standardized to 1.0 for controls (relative intensity). Isoprenaline (150 mg/kg s.c., given once (at time 0) or twice, at 0 and 24 h). First challenge for initial infarction induction, assessed at 24 h thereafter; then, at 24 h, the next application (isoprenaline given twice, at 0 and 24 h) for reinfarction induction (assessment at the end of the subsequent 24 h period). Medication (/kg, i.p.) (BPC 157 (10 µg) or saline (5 mL) (control)) at 30 min before isoprenaline. Means ± S.D. vs. control, * at least p < 0.05, 10 rats per each experimental group.

Experimental Group	eNOS/GAPDH		iNOS/GAPDH		COX-2/GAPDH	
	24 h after First Isoprenaline Challenge	48 h after First Isoprenaline Challenge	24 h after First Isoprenaline Challenge	48 h after First Isoprenaline Challenge	24 h after First Isoprenaline Challenge	48 h after First Isoprenaline Challenge
Saline 5 mL/kg i.p. + isoprenaline 150 mg/kg s.c.	1.23 ± 0.3	1.15 ± 0.1	1.05 ± 0.1	1.02 ± 0.1	1.05 ± 0.08	0.83 ± 0.1
BPC 157 10 µg/kg i.p. + isoprenaline 150 mg/kg s.c.	0.97 ± 0.2 *	1.06 ± 0.1	1.10 ± 0.2	1.06 ± 0.2	0.53 ± 0.08 *	0.93 ± 0.1

## Data Availability

The data presented in this study are available on request from the corresponding author.

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
