# Peer review of "Stable Gastric Pentadecapeptide BPC 157 May Counteract Myocardial Infarction Induced by Isoprenaline in Rats"

_biomedicines, 2022, doi:10.3390/biomedicines10020265_

Round 1

Reviewer 1 Report

Stable gastric pentadecapeptide BPC 157 may counteract myocardial infarction induced by isoprenaline in rats by Barisic I et al., is an interesting manuscript. The authors have clearly explained the protective role of BPC157 in countering MI induced by isoprenaline. For understanding their clinical implications, it is very important to understand the pharmacological and toxicological profile of this agent. The following experiments should be performed in the revised version. These experiments will really improve  the quality of the manuscript.

1) In the control animals it is very important to perform the dose dependent effects- on cellular toxicity( BPC 157 (10 ng/kg, 10 µg/kg i.p.)).

a) measure caspase activation, specifically caspase - 3 in control animals treated with BPC

b) measure LDH assay following BPC administration

c) measure markers of necrosis in BPC administered animals and compared to control animals.

2) Include a summary diagram depicting the mechanism of how BPC is bring about the cardiac protection.

3) Please perform a spell check in the revised version of the manuscript.

Author Response

Reviewer 1

Stable gastric pentadecapeptide BPC 157 may counteract myocardial infarction induced by isoprenaline in rats by Barisic I et al., is an interesting manuscript. The authors have clearly explained the protective role of BPC157 in countering MI induced by isoprenaline. For understanding their clinical implications, it is very important to understand the pharmacological and toxicological profile of this agent. The following experiments should be performed in the revised version. These experiments will really improve  the quality of the manuscript.

Acknowledged. To improve the understanding, we performed additional experiments, providing the evidence that isoprenaline may induce a huge noxious syndrome, occlusion-like syndrome, Early multiorgan failure, thrombosis, intracranial (superior sagittal sinus) hypertension, portal and caval hypertension, and aortal hypotension, full presentation (i.e. giant T-vawe) may be  decisive for further isoprenaline-lesion development, comparable to the syndrome seeable with major intoxication (alcohol, lithium) or maintained intra-abdominal hypertension that BPC 157 may fully counteract due to the activation of the collateral pathways, i.e. azygos vein shunt, providing direct delivery from inferior caval vein to superior caval vein, and reorganize blood flow to counteract induced vascular failure. This early syndrome may be likely decisive for further isoprenaline noxious course. To this point see our recent manuscripts, i.e. 

Biomedicines. 2021 Oct 20;9(11):1506. doi: 10.3390/biomedicines9111506 Biomedicines. 2021 Oct 20;9(11):1506. doi: 10.3390/biomedicines9111506 Biomedicines. 2021 Aug 17;9(8):1029. doi: 10.3390/biomedicines9081029 Biomedicines. 2021 Jul 8;9(7):792. doi: 10.3390/biomedicines9070792 Biomedicines. 2021 Jun 28;9(7):744. doi: 10.3390/biomedicines9070744. Biomedicines. 2021 May 26;9(6):609. doi: 10.3390/biomedicines9060609

In particular, the point of the toxicological profile is covered in the last concluding paragraph (LD1 was not achieved), a finding confirmed by an other study as well.

1) In the control animals it is very important to perform the dose dependent effects- on cellular toxicity( BPC 157 (10 ng/kg, 10 µg/kg i.p.)).

In particular, the point of the toxicological profile is covered by the high range of the application of BPC 157 µg – ng. Besides, we should mention again the previously emphasized evidence (LD1 was not achieved, a finding confirmed by an other study as well)

  1. a) measure caspase activation, specifically caspase - 3 in control animals treated with BPC
  2. b) measure LDH assay following BPC administration
  3. c) measure markers of necrosis in BPC administered animals and compared to control animals.

a-c, LDH and necrosis markers (CK, CK-MB, cTnT) had been already included (previous Figures 1 and 2 ). These points were now even more emphasized. Unfortunately, due to technical problems, we are unable to follow capsase activation.

2) Include a summary diagram depicting the mechanism of how BPC is bring about the cardiac protection.

Acknowledged. Summary diagram depicting the mechanism of how BPC 157 brings about the cardiac protection is now included (see Figure 20).

3) Please perform a spell check in the revised version of the manuscript.

Acknowledged.

Reviewer 2 Report

The authors investigated the putative cardioprotective effect of BPC 157, a gastral pentadecapepdide, in a rat ischemia model using both a prophylactic or therapeutic approach. The authors were able to demonstrate structurally-morphologically as well as functionally a protective effect of BPC 157. Here, the authors show an interaction with the NO signalling cascade in further pharmacological approaches that at least correlates to the protective effect of the peptide.  Overall, this is a profound in vivo data set that seems valid to me. The authors discuss their results in detail and in a comprehensible manner.

  • Since a cardioprotective effect of BPC 157 has already been shown in other models, it is unclear to me, however, why the authors present here another study with another stress-induced infarct model. What is not addressed here is a clear mechanistic effect of the peptide. It remains unclear at the cellular level how and where BPC 157 has a direct protective effect. An analysis, e.g. with cardiomyocytes, was not carried out here and thus the exact role or possible direct interaction in the NO signalling pathway remains unclear.
  • It should be emphasised very positively that the authors have implemented various controls, which they present transparently. Overall, however, the presentation in its current form with 12 different groups is very confusing and not comprehensible for the reader. The labelling of the groups was distributed over the two graphs in time regarding the quantitative data, which I do not consider appropriate. I therefore recommend simplifying the presentation and presenting some of the controls as separate graphs in a supplement.
  • Why were two different isoprenaline concentrations used?
  • Ad figure 4: Does such a severe remodelling already take place within 24 h in this model as can be seen in figure B?
  • In figure 5, the first two histological figures are unclear to me. The legend is difficult to understand here.
  • Tables 2-5 in their current form are also very complex, making it difficult to see the differences between the various groups. I also recommend a flow chart that summarises the methodology in an overview.
  • Ad figure 10: Why was the number of groups reduced here?
  • The supposed therapeutic cardiological relevance or innovation of BPC 157 should be discussed in more depth in the introduction or discussion.
  • Please discuss the prophylactic or therapeutic approach more in detail.
  • All quantitative data should be presented in form of scatter plot graphs so that the reader can better see the variance in the data.

Author Response

Reviewer 2

The authors investigated the putative cardioprotective effect of BPC 157, a gastral pentadecapepdide, in a rat ischemia model using both a prophylactic or therapeutic approach. The authors were able to demonstrate structurally-morphologically as well as functionally a protective effect of BPC 157. Here, the authors show an interaction with the NO signalling cascade in further pharmacological approaches that at least correlates to the protective effect of the peptide.  Overall, this is a profound in vivo data set that seems valid to me. The authors discuss their results in detail and in a comprehensible manner.

  • Since a cardioprotective effect of BPC 157 has already been shown in other models, it is unclear to me, however, why the authors present here another study with another stress-induced infarct model. What is not addressed here is a clear mechanistic effect of the peptide. It remains unclear at the cellular level how and where BPC 157 has a direct protective effect. An analysis, e.g. with cardiomyocytes, was not carried out here and thus the exact role or possible direct interaction in the NO signalling pathway remains unclear.

Acknowledged. The specific point of isoprenaline infarct model is elaborated and  explained in the Introduction, paragraph 1, paragraph 2, and paragraph 3.

Isoprenaline myocardial infarction in rats is known to be a rapid, simple and non-invasive method, producing myocardial damage similar to that seen in acute myocardial infarction in humans [1]. We hypothesize that the stable gastric pentadecapeptide BPC 157  (for review see, i.e., [2-4]) may be useful peptide therapy against isoprenaline-induced myocardial infarction, and reinfarction as well as against isoprenaline-induced antecedent early noxious effects so far less investigated.

Considering the particular points of the isoprenaline application [1], this may be because BPC 157 is known for the particular beneficial effect in the congestive heart failure [5] and in many severe arrhythmias models [6-11] and  therapy effect in the pulmonary hypertension studies [12,13]. Moreover, there is BPC 157 therapy rapid effect on heart disturbances in the studies of the permanent occlusion of major vessels-induced occlusion syndromes [14-20], and “occlusion-like” syndrome induced by severe intoxication (alcohol, lithium) [21,22]  and intra-abdominal hypertension [23], recovered with the activation of the collateral pathways to compensate vascular failure [14-23],  and compelling evidence that BPC 157 therapy realized both prophylactic and curative effect. This beneficial effect may be likely extended to the attenuation of the isoprenaline-induced noxious course. This may be also because  catecholamines might represent a common pathway in the evolution of myocardial changes in humans who develop myocardial lesions without narrowing or obstruction of coronary arteries [24].

For illustration, an activated azygos vein as a rescuing pathway, avoiding both the lung and liver,  combines the inferior caval vein and superior caval vein via direct blood delivery. Thus, activated azygos vein shunt could reorganize blood flow and instantly attenuate the consequences of maintained occlusion-induced vascular failure, both peripherally and centrally [14-20]. With major vascular occlusions, there were, at the periphery, the leading role of the rapid heart disturbances, and thereby, the heart, lung, liver, kidney and gastrointestinal lesions,  inferior and superior vena caval congestion; azygos vein failure, portal and caval hypertension and aortal hypotension [14-20]. Centrally, there were the intracranial (superior sagittal sinus) hypertension; brain swelling and severe lesions [14-20]. Widespread thrombosis occurred peripherally and centrally (note, BPC 157 maintained thrombocytes function (without interference with coagulation) [25-27], prevented in veins and arteries thrombosis formation, and abrogated already advanced thrombosis [14-23,28]).  A shared major “occlusion-like” syndrome, which was also antagonized by BPC 157 therapy, occurred after intragastric application of absolute alcohol or intraperitoneal application of lithium-overdose or with maintained severe intra-abdominal hypertension, grade III and grade IV [21-23]. Thus, the evidenced  leading role of the rapid heart disturbances and general vascular failure in the major vessels-induced occlusive syndromes [14-20] and in the severe intoxication (i.e. alcohol, lithium) [21,22] and maintained intra-abdominal hypertension [23] suggest that an application of isoprenaline  by its own can induce rapidly an own “occlusion-like”-syndrome, an early disturbance decisive for further isoprenaline-myocardial lesions progression. This particular early noxious effect of isoprenaline was not investigated so far, but it can be likely antagonized by BPC 157 therapy application.

Considering NO-system question, the specific point of BPC 157 and NO-system is fully explained in the last paragraph of Introduction. “....in the isoprenaline-rats, the implementation of NO-system-blockade (L-NAME), NO-system-over-stimulation (L-arginine) and NO-system immobilization (L-NAME+L-arginine) would fully consider the role of NO-system as an endogenous cardioprotectant [38]“  

Finally, isoprenaline-rats would receive pentadecapeptide BPC 157, L-NAME, L-arginine, alone and/or in combination. Further assessment includes eNOS, iNOS, COX-2 mRNA levels and lipid peroxidation in infarcted heart (known to be up-regulated in isoprenaline myocardial infarction [38,39]). Namely, isoprenaline interacts with NO-system and its blockade [38,41-43]. BPC 157/NO-system relations illustrate the evidence that  BPC 157 is known to affect several molecular pathways [30,44-52], in particular, having a modulatory effects on NO-system and prostaglandins-system [53,54] and on vasomotor tone and the activation of Src-Caveolin-1-eNOS pathway [45]. Consequently, BPC 157 induced the NO-release of its own [55,56], and  has large interaction with NO-system in various models and species [53], and the counteraction of the adverse effects of either NOS-blockade (L-NAME) or NOS-substrate L-arginine application (i.e. opposed hypertension and pro-thrombotic effect (L-NAME) [26,55], and hypotension and anti-thrombotic (L-arginine) effect [26,55]).  Further, in the isoprenaline-rats, the implementation of NO-system-blockade (L-NAME), NO-system-over-stimulation (L-arginine) and NO-system immobilization (L-NAME+L-arginine) would fully consider the role of NO-system as an endogenous cardioprotectant [38] (i.e., in rats and mice doxorubicine-congestive heart failure therapy with BPC 157 resulted in normalization of NO-system functioning since induced normalization of the increased endothelin-1 values [5]).

  • It should be emphasised very positively that the authors have implemented various controls, which they present transparently. Overall, however, the presentation in its current form with 12 different groups is very confusing and not comprehensible for the reader. The labelling of the groups was distributed over the two graphs in time regarding the quantitative data, which I do not consider appropriate. I therefore recommend simplifying the presentation and presenting some of the controls as separate graphs in a supplement.

Acknowledged. All Figures are completely redesigned and hopefully, much better organized and Figures legends completely rewritten.

  • Why were two different isoprenaline concentrations used?

Acknowledged. Use of the two isoprenaline doses seems to be along with the possibility that dose would support each other effect.

  • Ad figure 4: Does such a severe remodelling already take place within 24 h in this model as can be seen in figure B?
  • In figure 5, the first two histological figures are unclear to me. The legend is difficult to understand here.

Acknowledged. Improved and corrected.

  • Tables 2-5 in their current form are also very complex, making it difficult to see the differences between the various groups. I also recommend a flow chart that summarises the methodology in an overview.

Acknowledged. Improved and corrected. A summary figure is also included.

  • Ad figure 10: Why was the number of groups reduced here?

Acknowledged. The focus was on the effect of BPC 157.

  • The supposed therapeutic cardiological relevance or innovation of BPC 157 should be discussed in more depth in the introduction or discussion.

Acknowledged. These were fully discussed.

  • Please discuss the prophylactic or therapeutic approach more in detail.

Acknowledged. These were fully discussed.

  • All quantitative data should be presented in form of scatter plot graphs so that the reader can better see the variance in the data.

Acknowledged. As mentioned, all of the Figures are corrected, hopefully, providing an adequate clarity for the readers.  

Round 2

Reviewer 2 Report

The manuscript shows a clearly optimised form in terms of structure and content. The authors have sufficiently and constructively addressed my criticisms. I still recommend the exact specification of the statistical methodology in the figure legends as well as the n-number used. In Figures 1 and 13, the significance is partially unclear to me because no scatter is shown here.

Author Response

Dear Editor,

Re: Biomedicines 1345362

Thank you very much for your kind letter and acceptance of the manuscript providing that minor corrections will be done.

Considering the comment made by the reviewer, the number of the animals which was indicated in 2.1. section, is now included in the each of the Figures.

Besides, a final improvement is made in the Figure 20 and Figure 20 legend – to better summarized the isoprenaline background and BPC 157 therapy effect.

Once again, thank you very much for your kindness and assistance.

Sincerely

Predrag Sikiric, MD, PhD

Professor